# Asymmetric clustering of centrosomes defines the early evolution of tetraploid cells

**Nicolaas C Baudoin, Joshua M Nicholson, Kimberly Soto, Olga Martin, Jing Chen\*, Daniela Cimini\***

Department of Biological Sciences and Fralin Life Sciences Institute, Virginia Tech, Blacksburg, United States

**Abstract** Tetraploidy has long been of interest to both cell and cancer biologists, partly because of its documented role in tumorigenesis. A common model proposes that the extra centrosomes that are typically acquired during tetraploidization are responsible for driving tumorigenesis. However, tetraploid cells evolved in culture have been shown to lack extra centrosomes. This observation raises questions about how tetraploid cells evolve and more specifically about the mechanisms(s) underlying centrosome loss. Here, using a combination of fixed cell analysis, live cell imaging, and mathematical modeling, we show that populations of newly formed tetraploid cells rapidly evolve in vitro to retain a near-tetraploid chromosome number while losing the extra centrosomes gained at the time of tetraploidization. This appears to happen through a process of natural selection in which tetraploid cells that inherit a single centrosome during a bipolar division with asymmetric centrosome clustering are favored for long-term survival.

## Introduction

Organismal polyploidy is confined to certain taxa, but many species across the tree of life are thought to have had polyploid ancestors at some point in their evolutionary history (*Comai, 2005*; *Smith and Keinath, 2015*; *Smith et al., 2013*; *Wolfe, 2001*) and polyploidy is thought to contribute to speciation and evolution (*Bennett, 2004*; *Otto and Whitton, 2000*).

In vertebrates, organismal polyploidy is rare, and among mammals it has only been described in a single species (*Gallardo et al., 1999*). However, within individual diploid mammals, some tissues physiologically develop to have a higher ploidy than the majority of somatic cells (*Nagl, 1990*; *Orr-Weaver, 2015*; *Øvrebø and Edgar, 2018*; *Schoenfelder and Fox, 2015*). Polyploidy can also occur outside of the context of normal development, and is linked with both pathology (particularly cancer [*Ganem et al., 2007*]) and aging (*Tanaka et al., 2015*). Tetraploid cells are commonly found in pre-malignant lesions and tumors at different stages (*Davoli and de Lange, 2011*; *Galipeau et al., 1996*; *Olaharski et al., 2006*). Furthermore, meta-analysis of catalogued tumor genomes has provided evidence that close to 40% of all cancers – even those that were not tetraploid at the time of sampling – had a tetraploid intermediate stage at some point during tumor evolution (*Zack et al., 2013*). Consistent with this, several studies have shown a direct, causative link between tetraploidy and tumorigenesis (*Fujiwara et al., 2005*; *Nguyen et al., 2009*).

In proliferating cells, tetraploidy can arise via abnormal cell cycle events, including cytokinesis failure, cell fusion, endoreduplication, and mitotic slippage (*Davoli and de Lange, 2011*; *Dikovskaya et al., 2007*; *Edgar and Orr-Weaver, 2001*; *Larsson et al., 2008*; *Rieder, 2011*). Most of these events result in the concomitant acquisition of extra centrosomes along with genome duplication. Importantly, both tetraploidy and extra centrosomes have been shown to trigger a p53-dependent arrest in some experimental systems (*Andreassen et al., 2001*; *Fava et al., 2017*). In

\*For correspondence:
chenjing@vt.edu (JC);
cimini@vt.edu (DC)

**Competing interests:** The authors declare that no competing interests exist.

experimental systems in which such an arrest does not occur, extra centrosomes have been shown to promote chromosomal instability (*Ganem et al., 2009*; *Silkworth et al., 2009*) and invasive/migratory behavior (*Godinho et al., 2014*). Recent studies have also shown that extra centrosomes promote and in some cases are sufficient to drive tumorigenesis in vivo (*Levine et al., 2017*; *Serçin et al., 2016*).

Based on these studies, it has been speculated that the extra centrosomes emerging as a result of tetraploidization may drive chromosomal instability and, in turn, tumorigenesis (*Storchova and Pellman, 2004*). However, it was previously reported that cytokinesis failure does not result in stable centrosome amplification in a cell population (*Krzywicka-Racka and Sluder, 2011*). Moreover, anecdotal reports (*Ganem et al., 2009*; *Godinho et al., 2014*; *Kuznetsova et al., 2015*; *Potapova et al., 2016*) have indicated that clones of tetraploid or near-tetraploid cells displayed normal centrosome numbers. This suggests that our understanding of the evolution of tetraploid cells is incomplete and how centrosome and chromosome numbers evolve after tetraploidization needs to be revisited. To address this problem, we studied the time period immediately following cytokinesis failure and investigated how centrosome and chromosome numbers change in newly formed tetraploid cells. Following the observation that the number of centrosomes, but not chromosomes, rapidly returns to normal, we combined computational and experimental approaches to identify a specific cellular mechanism that underlies the loss of extra centrosomes.

## Results

To investigate the early consequences of tetraploidy and the evolution of newly formed tetraploid cells, we induced cytokinesis failure by dihydrocytochalasin B (DCB) treatment for 20 hr (*Figure 1A*) in both DLD-1 (pseudodiploid colorectal cancer cells) and p53$^{-/-}$ hTERT-immortalized RPE-1 cells (*Izquierdo et al., 2014*) (hereafter referred to as RPE-1 p53$^{-/-}$; p53-null RPE-1 cells were used because the parental, p53-positive, cells display a G1 arrest after cytokinesis failure, as shown in *Ganem et al., 2014*). Cells generated by this method are referred to, throughout the paper, as 'newly formed tetraploid cells' (text) or '4N new' (figures).

### Newly formed tetraploid cells undergo diverse fates in their first mitotic division

To determine the fate of the first tetraploid mitosis, we performed live-cell phase contrast microscopy for 24 hr following DCB washout (*Figure 1A*; note, newly formed tetraploid cells can easily be identified by the presence of two nuclei). We found that multipolar divisions were frequent in both cell types (*Figure 1B,C*), consistent with the acquisition of extra centrosomes upon cytokinesis failure and with the ability of extra centrosomes to promote formation of multipolar mitotic spindles. We confirmed the high rates of multipolar divisions by analyzing ana-/telophase cells immunostained for α–tubulin and centrin (*Figure 1—figure supplement 1A,B*). This fixed-cell analysis and analysis of live RPE-1 p53$^{-/-}$ cells with GFP-tagged centrin (*Figure 1—figure supplement 1C,D*) also confirmed that spindle poles always contained centrosomes (i.e., two centrin dots), as acentrosomal poles were never observed in multipolar ana-/telophase cells. Furthermore, all of the live binucleate cells that we observed contained supernumerary centrosomes (*Figure 1—figure supplement 1D*). These centrosomes were duplicated prior to mitosis and were never lost/extruded during mitosis (*Figure 1—figure supplement 1D*). The observation that only ∼20–30% of newly formed tetraploid cells underwent bipolar anaphase indicates that centrosome clustering is not prevalent in newly formed tetraploid cells. However, when we followed these multipolar mitoses through cytokinesis, we found that tripolar or tetrapolar anaphases did not always generate three or four daughter cells, respectively (*Figure 1D–F*). Instead, the DNA corresponding to two or more anaphase poles was often enclosed in a single daughter cell, giving rise to binucleated or, rarely, trinucleated daughter cells (*Figure 1D–F*), consistent with previous observations (*Chen et al., 2016*; *Wheatley and Wang, 1996*).

### Highly aneuploid cells form early in the evolution of tetraploid cells, but quickly disappear from the population

We next investigated how these early cell divisions after tetraploidization may impact chromosome numbers in the proliferating cell population. To this end, we carried out a time-course experiment in

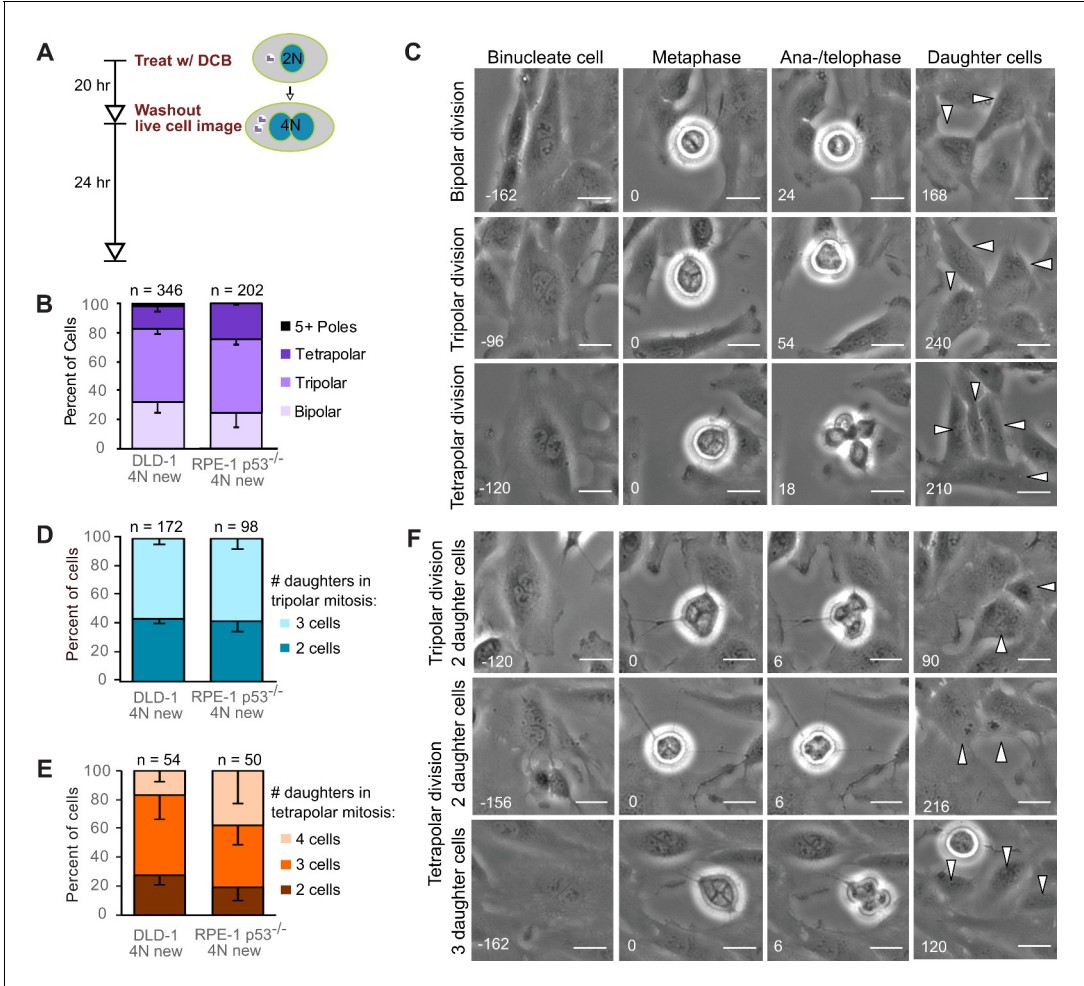

**Figure 1.** Newly formed tetraploid cells undergo diverse fates in their first mitotic division. (**A**) Experimental design for generation of newly formed tetraploid cells, followed by live cell imaging. DCB, dihydrocytochalasin B. (**B**) Quantification of the types of division observed in the first cell division of newly formed tetraploid cells; characterization was performed at ana-/telophase. (**C**) Examples of bipolar (top), tripolar (middle), and tetrapolar (bottom) divisions. (**D–E**) Quantification of incomplete cytokinesis in tripolar (**D**) and tetrapolar (**E**) divisions; n-values represent the number of tripolar and tetrapolar mitoses that displayed incomplete cytokinesis out of all the cells analyzed in **B**). (**F**) Examples of multipolar divisions with incomplete cytokinesis such that multiple anaphase poles are incorporated into a single daughter cell. Error bars in all graphs represent S.E.M. from three independent experiments. All scale bars, 25 μm. Arrowheads in all images point to individual daughter cells.

The online version of this article includes the following source data and figure supplement(s) for figure 1:

**Source data 1.** Source data for *Figure 1B,D,E*.
**Figure supplement 1.** Centrosomes are present at each spindle pole in multipolar ana-/telophase cells.
**Figure supplement 1—source data 1.** Source data for *Figure 1—figure supplement 1B,D*.

which we performed chromosome counting in the cell population after the 20 hr DCB treatment and every two days thereafter for a 12 day period (*Figure 2A–B*). Immediately after drug washout, we observed a tetraploid fraction corresponding to approximately 80% and 60% of the population for DLD-1 and RPE-1 p53$^{-/-}$ cells, respectively (*Figure 2C–E,G*, day 0). Two days after DCB washout, we observed high frequencies of cells with chromosome counts in the hypotetraploid/hyperdiploid range (*Figure 2B*, middle panel; *Figure 2C–E,G*). It is conceivable that these highly aneuploid cells may originate from the multipolar mitoses we observed in our live-cell imaging experiments. Indeed, by quantifying DNA fluorescence of separated chromosome masses in fixed ana-/telophase cells (*Figure 2—figure supplement 1A*), we observed that while chromosome distribution to the daughter cells was balanced in bipolar divisions (*Figure 2—figure supplement 1B–D*), it greatly deviated from an equal distribution in multipolar mitoses (*Figure 2—figure supplement 1B–D*), indicating

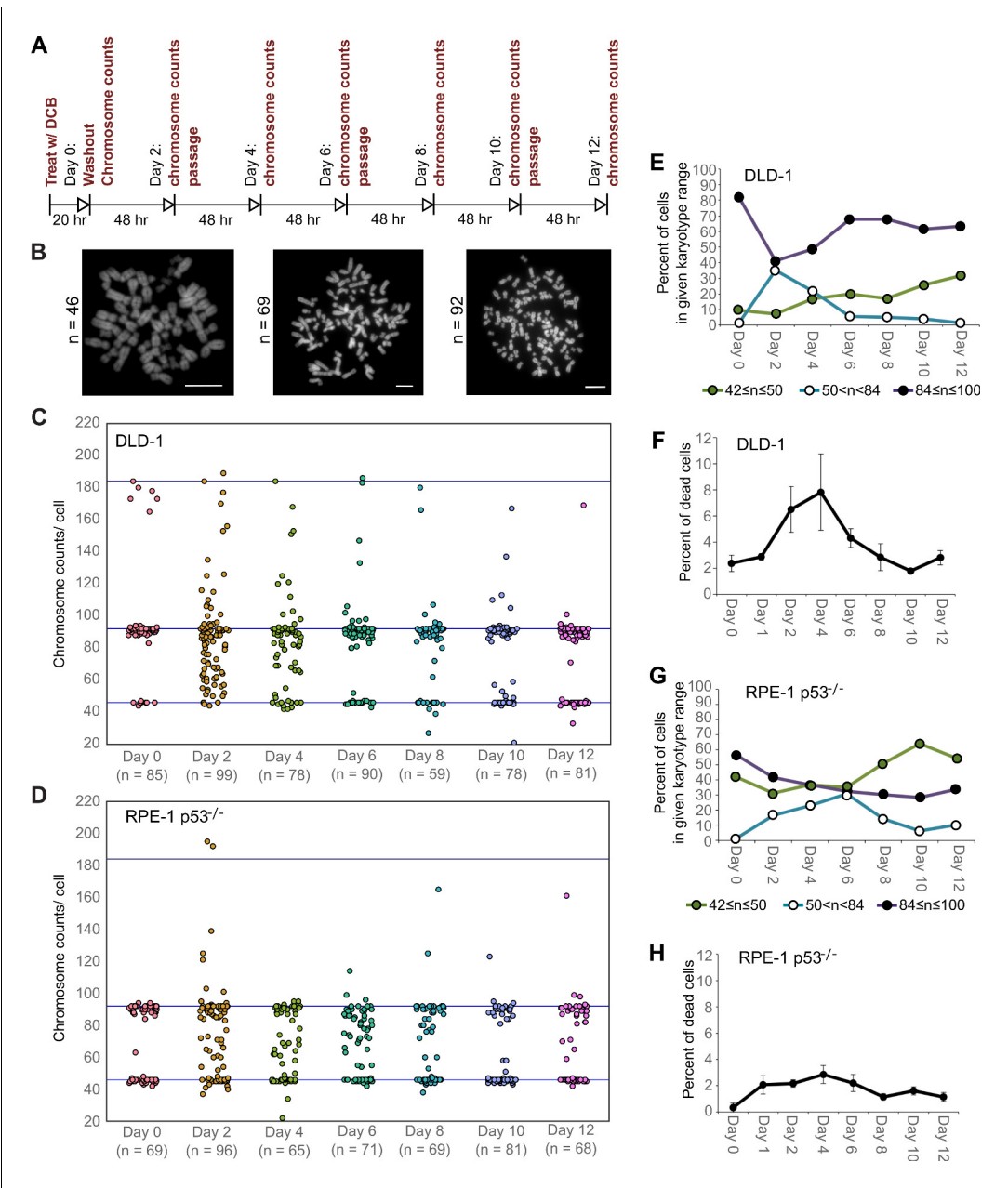

**Figure 2.** High degrees of aneuploidy appear and rapidly disappear following tetraploidization. (**A**) Experimental design for time course experiments to analyze chromosome number evolution over a 12 day time period after experimental induction of tetraploidization by dihydrocytochalasin B (DCB). (**B**) Example chromosome spreads from cells with diploid (left), highly aneuploid (middle), and tetraploid (right) chromosome numbers. Scale bars, 10 μm. (**C–D**) 12 day time course analysis of chromosome numbers in DLD-1 (**C**) and RPE-1 p53[-/-] (**D**) cell populations after induction of tetraploidization. (**E**) Quantification (from the data in **C**) of the fraction of cells that are near-diploid (green), highly aneuploid (blue/white), or near-tetraploid (purple/black). (**F**) Time course analysis of cell death in DLD-1 cell populations with newly formed tetraploid cells. (**G**) Quantification (from the data in **D**) of the fraction of cells that are near-diploid (green), highly aneuploid (blue/white), or near-tetraploid (purple/black). (**H**) Time course analysis of cell death in RPE-1 p53[-/-] cell populations with newly formed tetraploid cells. Chromosome number data were obtained from two independent experiments. Error bars in (**F, H**) represent S.E.M. from three independent experiments.

The online version of this article includes the following source data and figure supplement(s) for figure 2:

**Source data 1.** Source data for *Figure 2C–H*.

**Figure supplement 1.** DNA can be distributed unevenly to the three (tripolar) or four (tetrapolar) poles of multipolar divisions.

**Figure supplement 1—source data 1.** Source data for *Figure 2—figure supplement 1B,C*.

that the daughter cells in multipolar divisions inherit variable proportions of the genome. However, the fraction of cells with highly abnormal chromosome numbers rapidly decreased over the course of the 12 day experiment in both cell types and highly aneuploid cells were virtually eliminated from

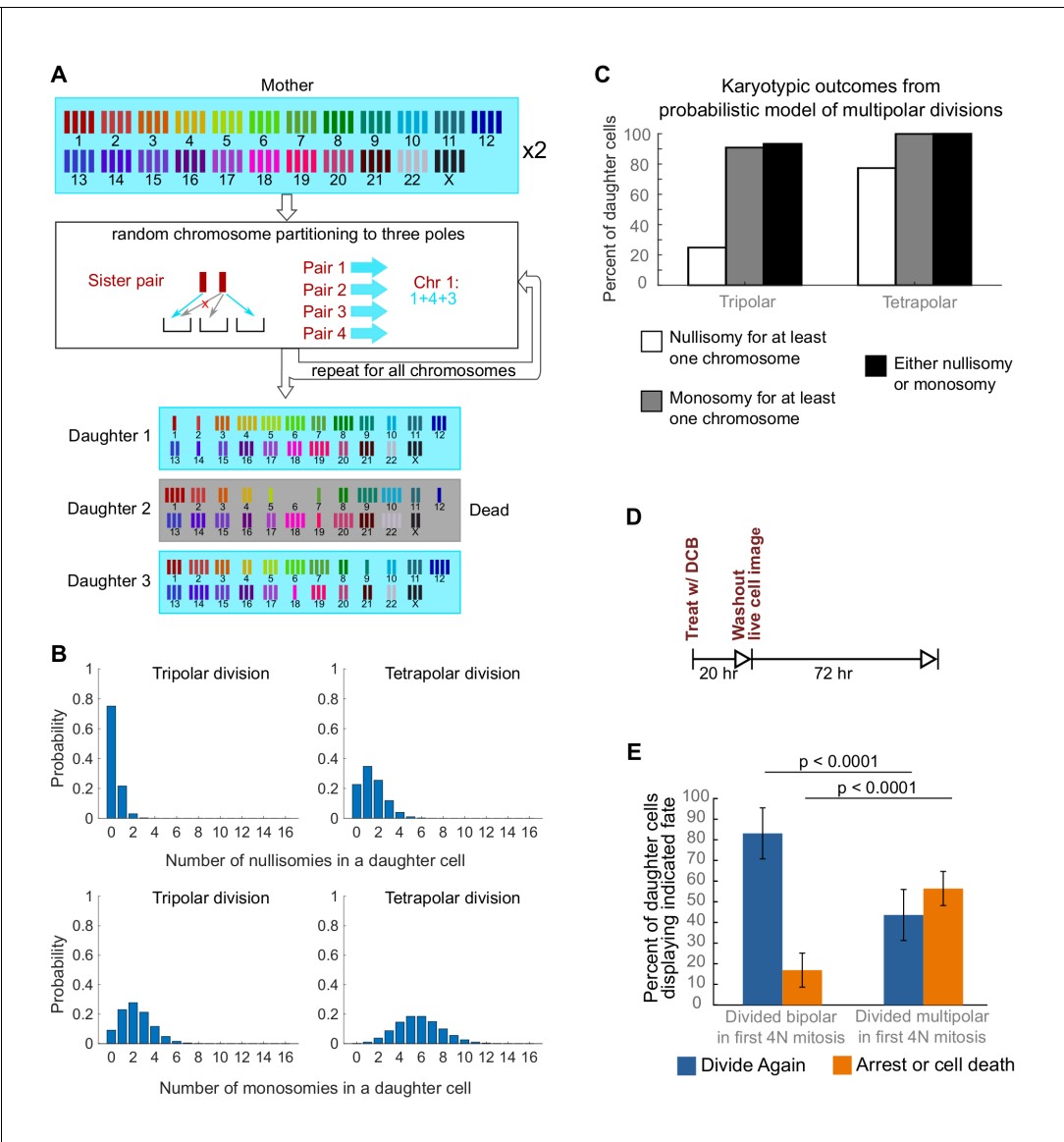

**Figure 3.** Daughters of multipolar divisions are likely to bear nullisomies or monosomies and are more likely to die or arrest over the subsequent 48 hr compared to the daughters of bipolar divisions. (A) Probabilistic model for random partitioning of chromosomes in multipolar divisions. An example is shown for how chromosome partitioning occurs in the model in a tripolar division of a tetraploid cell. Chromosomes are randomly partitioned to three poles (cyan small arrows), with sister chromatids to different poles (red cross eliminating the case with two sisters to the same pole). Daughter 2 is dead due to nullisomy of Chromosome 6. (B) Model predicted probability distributions of the number of nullisomies (top) or monosomies (bottom) in a daughter cell from tripolar (left) or tetrapolar (right) division of a tetraploid mother cell. Analytic formulas of the probabilities are given in Sections 1.1 and 1.2 of the Modeling Methods. (C) Model predicted probabilities of nullisomy and/or monosomy for at least one chromosome in daughter cells of tetraploid cells undergoing multipolar division. Note, there may be nullisomy for one chromosome and monosomy for another chromosome within the same cell. (D) Experimental design for live cell imaging to analyze the first two mitotic divisions of newly formed tetraploid cells. (E) Quantification of the fate of daughter cells derived from either a bipolar or a multipolar mitosis during the first tetraploid cell division. Error bars represent weighted S.E. M. from three independent experiments (weighted based on the number of cells analyzed in each experiment).

The online version of this article includes the following source data and figure supplement(s) for figure 3:

**Source data 1.** Source data for *Figure 3E* and *Figure 3—figure supplement 1*.

**Figure supplement 1.** Fates of newly formed tetraploid cells tracked for the first 72 hr post-cytokinesis failure.

the DLD-1 population by day 12, leaving sub-populations of near-diploid cells (presumably derived from cells that did not respond to the initial DCB treatment) and near-tetraploid cells (*Figure 2B,C–E,G*). The appearance and loss of highly aneuploid cells from the RPE-1 p53$^{-/-}$ cell population was delayed compared to the DLD-1 cell population (*Figure 2G*), possibly suggesting that newly formed tetraploid RPE-1 p53$^{-/-}$ cells display lower proliferation and death rates. Indeed, the appearance and disappearance of highly aneuploid cells corresponded to an increase followed by a decline in the fraction of dead cells between day 2 and 6 in both cell types (*Figure 2F,H*). However, cell death rates were lower in RPE-1 p53$^{-/-}$ compared to DLD-1 cells (*Figure 2F* vs. *Figure 2H*), possibly explaining the delay in both elimination of highly aneuploid cells and proliferation of ~4N cells in the RPE-1 p53$^{-/-}$ cell population (*Figure 2G*).

To investigate a possible cause for the disappearance of cells with highly aneuploid chromosome counts, we built a probabilistic model to evaluate the karyotypic outcomes of multipolar divisions (see Materials and Methods and *Figure 3A*). The model predicted that daughter nuclei emerging from multipolar divisions in tetraploid cells were very likely to bear a monosomy or nullisomy for at least one chromosome (*Figure 3B,C*). Because nullisomic cells and cells with certain monosomies are expected to be unable to proliferate further, the daughters of multipolar divisions would be expected to display lower proliferation rates than the daughters of bipolar divisions. This was confirmed in long-term time lapse microscopy experiments in newly formed tetraploid DLD-1 cells (*Figure 3D*), which showed that daughter cells produced by multipolar divisions were more likely to die or arrest compared to cells produced by bipolar divisions (*Figure 3E*, *Figure 3—figure supplement 1*). Previous studies have shown that diploid and near-triploid cells undergoing multipolar divisions produce daughter cells that arrest or die (*Ganem et al., 2009*; *Gisselsson et al., 2010*). Our data indicate that this is also the case for tetraploid cells undergoing multipolar divisions.

## Supernumerary centrosomes that arise through tetraploidization quickly disappear from the population

The observation that highly aneuploid cells make up only a very small fraction of the population by 12 days after tetraploidization suggests that the rate of multipolar divisions (which generate these cells) decreases over time. This could be due to either an increased ability of the extra centrosomes to cluster in the tetraploid cells or to elimination of the extra centrosomes. To explore this, we investigated if and how centrosome number varies over the same 12 day evolution period by analyzing cells immunostained for centrin immediately following cytokinesis failure and every two days thereafter (*Figure 4A*).

We performed this analysis in both mitotic and G1 cells and obtained similar results at all time points, except immediately following DCB washout ('Day 0' in *Figure 4C,D*). This discrepancy at day 0 could be explained by a delay in mitotic entry of newly formed 4N cells, particularly for the RPE-1 p53$^{-/-}$ cells. Despite this difference at day 0, the trend was clear: the fraction of the G1 cell population containing supernumerary centrioles after a 20 hr cytokinesis block was 90% and 87.3% in DLD-1 and RPE-1 p53$^{-/-}$ cells, respectively. However, this fraction rapidly diminished over the 12 day observation period, reaching frequencies that are close to the frequencies of cells with supernumerary centrioles in the parental populations. Moreover, the fraction of cells with supernumerary centrioles at day 12 (*Figure 4C–D*) was substantially smaller than the fraction of cells with ~4N chromosome number (16.3% and 13.3% vs. 63% and 33%, respectively, in DLD-1 and RPE-1 p53$^{-/-}$; compare 'Day 12' data from *Figure 4C,D* and *Figure 2E,G*). Indeed, statistical analysis showed a highly significant difference between the number of cells with ~4N chromosome number and cells with extra centrosomes at day 12 for both DLD-1 and RPE-1 p53$^{-/-}$ cells (two-sided Chi square test, p<0.0001 for both cell lines), indicating that a large fraction of the tetraploid cells that are present 12 days post-cytokinesis failure have lost their extra centrosomes.

## Tetraploid cells can inherit a normal centrosome number through asymmetric centrosome clustering during cell division

We reasoned that one way in which tetraploid cells could regain a normal centrosome number while maintaining tetraploid chromosome numbers would be by asymmetrically clustering the centrosomes during formation of a bipolar spindle. As a result, one daughter cell would receive three

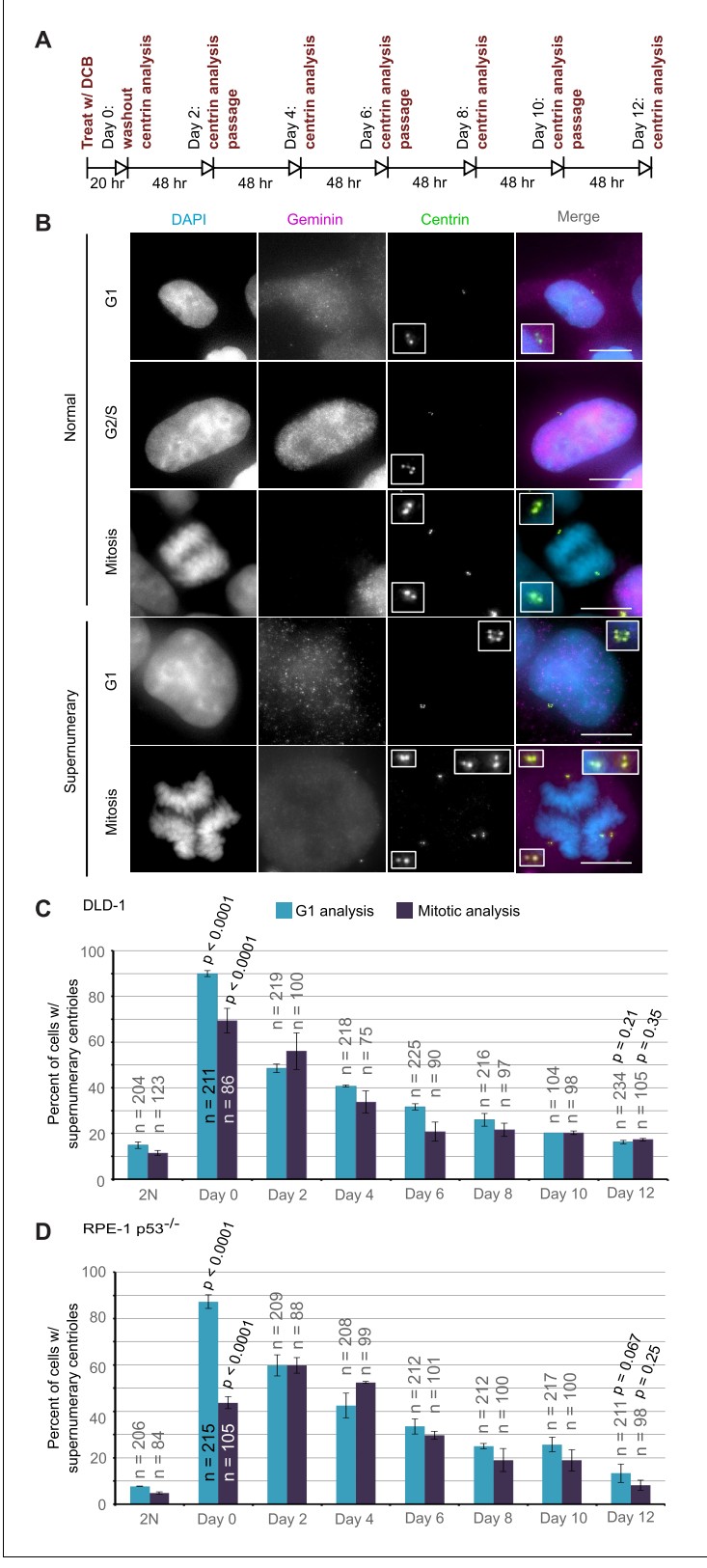

**Figure 4.** Extra centrosomes are rapidly lost from the cell population after tetraploidization. (**A**) Experimental design for time course experiments to analyze centrosome number in cell populations evolving over a 12 day period after induction of tetraploidization. DCB, dihydrocytochalasin B. (**B**) Examples of interphase and mitotic cells with normal centrosome number (top) or supernumerary (bottom) centrosomes. Scale bars, 10 μm. (**C–D**) 12

*Figure 4 continued on next page*

*Figure 4 continued*

day time course analysis of centrosome number in mitotic and G1 (cells negative for nuclear geminin staining) DLD-1 (**C**) and RPE-1 p53⁻/⁻ (**D**) cells after induction of tetraploidization. Centrosome number data are reported as mean ± S.E.M. from two independent experiments in which the total number of cells reported on each bar was analyzed. The reported p-values refer to comparison between individual data point and the corresponding data in the parental 2N cell line by two-sided Fisher's exact test.

The online version of this article includes the following source data for figure 4:

**Source data 1.** Source data for *Figure 4C–D*.

---

centrosomes and the other daughter would receive one centrosome, but both would receive a chromosome number ~4N.

To explore this possibility, we analyzed bipolar mitotic DLD-1 and RPE-1 p53⁻/⁻ cells fixed and immunostained for centrin immediately following washout of DCB (*Figure 5A–B*). We found nearly equal numbers of bipolar DLD-1 cells with symmetric vs. asymmetric centrosome clustering in late mitosis (metaphase, anaphase, or telophase), while bipolar RPE-1 p53⁻/⁻ showed a slight bias towards symmetric clustering of centrosomes (*Figure 5C*).

For asymmetric centrosome clustering to explain evolution of a tetraploid cell population with normal centrosome number, one would also have to assume that the daughter cell inheriting a single centrosome has a selective advantage over the daughter cell inheriting extra centrosomes (e.g., due to the likelihood of multipolar division in cells with extra centrosomes). To test this, we built a mathematical model based on this assumption (for model details, see Materials and Methods, *Figure 5—figure supplements 1–4*, and *Table 1*). We started with a simple model (Model I, *Figure 5—figure supplement 1A–C*) in which, initially, 87–90% of the cells have two centrosomes in G1 (four in S/G2/M), corresponding to the experimentally observed frequencies after DCB treatment. Cells in the model can divide in a multipolar or bipolar fashion, and bipolar divisions can occur with either symmetric or asymmetric centrosome clustering (*Figure 5—figure supplement 1B–C*) – all with probabilities that reflect those observed experimentally (see *Table 1* for details on which experimental data motivated various model parameters). The daughter cells from multipolar divisions have significantly reduced viability and are expected to be quickly eliminated by selection; based on this, in the model these cells were, for simplicity, assumed to die (*Figure 5—figure supplement 1A–C*). Cells inheriting a single centrosome were assumed to become stable cells that undergo bipolar divisions with high viability (*Figure 5—figure supplement 1B–C*). Cells inheriting two centrosomes would display the same fate as newly formed tetraploid cells, and cells inheriting three centrosomes were assumed to undergo multipolar division and consequently produce non-viable progeny (*Figure 5—figure supplement 1B–C*). Although this model (Model I) captured centrosome loss, it predicted centrosome loss over a much shorter time scale than was observed experimentally for either DLD-1 or RPE-1 p53⁻/⁻ cells (*Figure 5D–E*, orange line). The final fraction of cells with extra centrosomes predicted from the model was also substantially lower than what was experimentally observed (*Figure 5D–E*, orange line). Parameter optimization within a reasonable range could not solve this discrepancy (*Figure 5—figure supplement 3A*). In particular, the final steady-state fraction of cells with extra centrosomes was strongly dependent on the probability of cytokinesis failure in cells with normal centrosome number (which generates new cells with extra centrosomes) (*Figure 5—figure supplements 1D* and *3A*). For the experimentally quantified (*Nicholson et al., 2015*) probability (~2.5%) of spontaneous cytokinesis failure in DLD-1 cells, the steady-state fraction of cells with extra centrosomes cannot match the observed value.

We next considered the possibility that a sub-fraction of newly formed tetraploid cells may cluster their centrosomes more efficiently than other cells (Model II, *Figure 5—figure supplement 2*), which we herein dub 'super-clustering' (SC) cells. When such SC cells were included in the model (Model II, *Figure 5—figure supplement 2*), the model output was no longer constrained by the probability of cytokinesis failure (*Figure 5—figure supplement 3B*) and the final fraction of cells with extra centrosomes could match the experimentally observed values (*Figure 5D–E*, blue line). Moreover, our model results showed that these SC cells would persist in the population and therefore dominate the final population of cells with supernumerary centrosomes (*Figure 5F–G*).

Altogether, our modeling results show that asymmetric centrosome clustering, along with a selective advantage of cells that inherit a single centrosome, is sufficient to explain the loss of extra

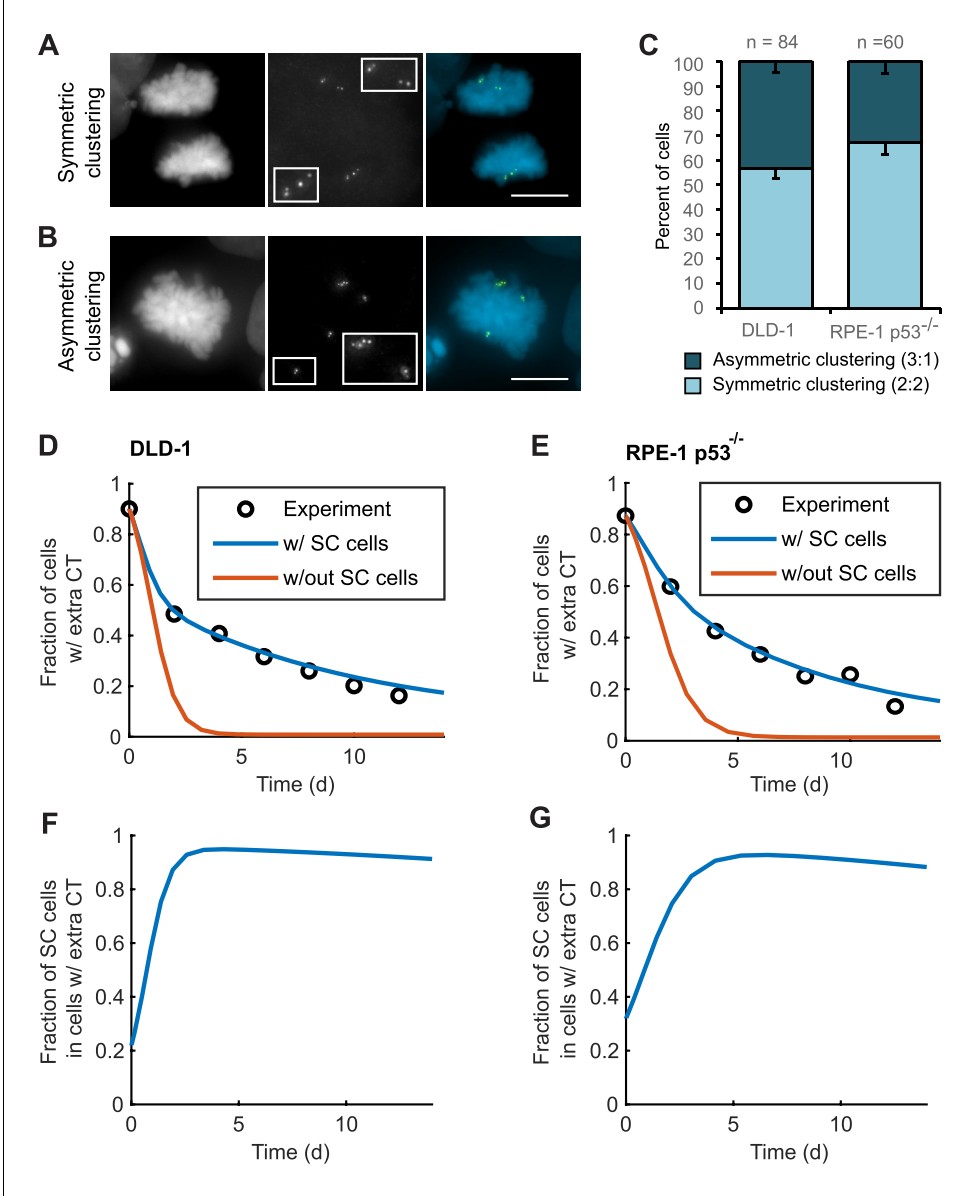

**Figure 5.** Asymmetric clustering of centrosomes in bipolar divisions can explain the formation of tetraploid cells with a normal centrosome number. (**A**) Example of anaphase cell with supernumerary centrosomes clustered symmetrically into a bipolar configuration, such that two centrosomes (four centrioles) are associated with each of the anaphase chromosome masses. (**B**) Example of a late prometaphase cell with supernumerary centrosomes clustered asymmetrically into a bipolar configuration, such that one centrosome (two centrioles) is at one side of the chromosome mass and three centrosomes (six centrioles) are on the other side of the chromosome mass. Scale bars, 10 µm. (**C**) Quantification of symmetric vs. asymmetric centrosome clustering in newly formed tetraploid DLD-1 and RPE-1 p53$^{-/-}$ mitotic cells with bipolar configuration. Data are reported as mean ± S.E.M. from at least three independent experiments. (**D–E**) Modeling results for centrosome evolution in DLD-1 (**D**) and RPE-1 p53$^{-/-}$ (**E**) cells based on Model I, without the added assumption that a fraction of cells clusters their extra centrosomes with high efficiency ('w/out SC cells,' orange) or Model II, with the added assumption that a subset of cells displays high centrosome clustering efficiency ('w/ SC cells,' blue). The modeling results are superimposed on the mean values of the experimental data (circles) from *Figure 4C–D*. When available, experimentally measured parameter values were used (see modeling methods for further details and *Table 1* for parameter values). (**F–G**) Fractions of cells with supernumerary centrosomes that are, over time, SC cells for DLD-1 (**F**) and RPE-1 p53$^{-/-}$ (**G**) cells based on Model II (with SC cells).

The online version of this article includes the following source data and figure supplement(s) for figure 5:

**Source data 1.** Source data for *Figure 5C*.

*Figure 5 continued on next page*

*Figure 5 continued*

**Figure supplement 1.** Scheme and parameter sensitivity analysis for Model I.
**Figure supplement 2.** Scheme and parameter sensitivity analysis for Model II.
**Figure supplement 3.** Final steady state fractions of cells with extra centrosomes are strongly constrained by the most sensitive parameters in both models.
**Figure supplement 4.** Fractions of each cell type in the total population approach steady state even though the total population size grows infinitely.

centrosomes in newly formed tetraploid cells, leading to the evolution of cell populations with tetraploid chromosome numbers but normal centrosome numbers (i.e., 1 centrosome, 2 centrioles in G1 tetraploid cells).

## Long-term live-cell imaging confirms that centrosome elimination and stable tetraploid cells arise via asymmetric centrosome clustering and natural selection

To directly observe the process of centrosome loss and test the model assumption that cells inheriting a single centrosome from a bipolar division are the most likely to keep proliferating, we performed live cell imaging experiments in DLD-1 and RPE-1 p53$^{-/-}$ cells expressing GFP-tagged centrin.

Because previous observations (*Ganem et al., 2009*; *Gisselsson et al., 2010*) and our own data (*Figure 3C,E*) indicated that the progeny of multipolar divisions display reduced viability, and since only bipolar or near-bipolar divisions are likely to generate the evolved (day 12) near-tetraploid cell population observed in our time-course experiment, we focused on fates of daughter cells arising from bipolar divisions. We imaged newly generated tetraploid (binucleate) cells by phase contrast microscopy for 24 hr, after which we determined the number of GFP-centrin dots present in the daughter cells arising from bipolar divisions. These cells were then imaged for an additional 48 hr by phase contrast microscopy (*Figure 6A*) to determine their fates in relation to the number of centrosomes they inherited. We found that cells that divided in a bipolar manner showed a mix of symmetric and asymmetric centrosome clustering without a strong preference for one mode (*Figure 6— figure supplement 1*), consistent with our fixed-cell data (*Figure 5A–C*; two-sided Fisher's exact test, p=0.8224 and p=0.2243 for fixed vs. live cell data in DLD-1 and RPE-1 p53$^{-/-}$, respectively). Cells that inherited a normal centrosome number (1 centrosome/2 centrioles) were significantly more likely than cells that inherited supernumerary centrosomes to divide in a bipolar manner in both DLD-1 and RPE-1 p53$^{-/-}$ cells (*Figure 6B–C*). In contrast, cells that inherited too many centrosomes went through a mix of fates, dominated by multipolar divisions, arrest, and cell death (*Figure 6—figure supplement 1*). These data, together with our mathematical modeling, strongly suggest that populations of stably dividing tetraploid cells containing a normal number of centrosomes can arise via asymmetric clustering of centrosomes (3:1) in bipolar mitoses and selective pressure against cells that inherit extra centrosomes.

In generating our Model II, we included the assumption that a fraction of cells with extra centrosomes had a very high efficiency of centrosome clustering ('SC cells'). This assumption was required to reproduce the observed evolution dynamics and final fraction of cells with extra centrioles, given the observed rate of cytokinesis failure. Based on this assumption, the model predicted that the fraction of SC cells rapidly increased over the first few days and that SC cells would make up about 90% of the remaining cell population with extra centrosomes at the end of the 12 day evolution period (*Figure 5F,G*). To test this model prediction, we analyzed fixed DLD-1 cells in ana-/telophase to determine the fractions of cells with extra centrioles that displayed bipolar vs. multipolar configurations. As predicted by our model, we found that approximately 90% of cells with extra centrioles displayed a bipolar configuration at day 12 compared with just 28% in newly formed tetraploid cells at Day 0 (*Figure 6D*). These results suggest that when extra centrosomes arise, they may only be retained when cells can cluster them efficiently, whereas cells that cannot cluster their centrosomes efficiently may disappear from the population.

**Table 1.** Model parameters for DLD-1 and RPE-1 p53$^{-/-}$ cells.

| Symbols | Description | DLD-1 | RPE-1 p53$^{-/-}$ | Range for data fitting (both models) | Reason/Source of information |
|---|---|---|---|---|---|
| $b_{C2}$ | Proliferation rate of $C_2$ and SC cells | 1.2 d$^{-1}$ | 0.94 d$^{-1}$ | 0.8~1.2 d$^{-1}$ | Range estimated from growth curves (not shown) |
| $b_{C4}$ | Proliferation rate of $C_4$ cells | 1 d$^{-1}$ | 0.6 d$^{-1}$ | 0.6~1 d$^{-1}$ | Range estimated from growth curves (not shown) |
| $q$ | Probability of bipolar division in $C_2$ cells | 0.975 | 0.975 | 0.975~1 | 1 - probability of cytokinesis failure (*Nicholson et al., 2015*) |
| $p$ | Probability that a $C_4$ cell undergoes bipolar division | 0.33 | 0.25 | Fixed | *Figure 1B* |
| $r$ | Probability of symmetric division in a bipolar division of $C_4$ cell | 0.5 | 0.7 | Fixed | *Figure 5C* |
| $fs$ | Probability that a $C_4$ cell deriving from a multipolar division of $C_4$ survives | 0.4 | 0.7 | Fixed | Inferred, *Figure 3—figure supplement 1* |
| $d_{C2}$ | Death rate of $C_2$ and SC cells | 0 | 0 | Fixed | Rates of spontaneous cell death are negligible in both cell lines |
| $d_{C4}$ | Death rate of $C_4$ cells | 0.5 d$^{-1}$ | 0.12 d$^{-1}$ | Fixed | Inferred, *Figure 2F, H* |
| $d_{C6}$ | Death rate of $C_6$ cells | 1.5 d$^{-1}$ | 1.5 d$^{-1}$ | Fixed | Comparable to rate of cell division, because $C_6$ progeny dies due to multipolar division |
| $v$ | Probability of getting SC cell from a cytokinesis failure event | 0.22 | 0.32 | 0~0.6 † | Range suggested by *Figure 3—figure supplement 1*; value obtained from data fitting (*Figure 5D, E*) |
| $r_S$ | Probability that an SC cell divides symmetrically | 0.93 | 0.90 | 0.5~1 † | Reason for range: the SC subpopulation likely sustains itself via symmetric divisions; value obtained from data fitting (*Figure 5D, E*) |

† Parameters that only apply to Model II.

## Discussion

### Newly formed tetraploid cells rapidly lose the extra centrosomes while maintaining the extra chromosomes

Here, we show, in two different cell lines, that populations of newly formed tetraploid cells rapidly evolve in vitro to retain a near-tetraploid chromosome number while losing the extra centrosomes gained at the time of tetraploidization. By combining fixed cell analysis, live cell imaging, and mathematical modeling, we show that this happens by a process of natural selection (*Figure 7*). Specifically, tetraploid cells that inherit a single centrosome during a bipolar division with asymmetric centrosome clustering are favored for long-term survival. Conversely, the majority of cells with extra centrosomes are eventually eliminated because of their high probability of undergoing multipolar division, which has a high likelihood of producing daughters with insufficient amounts of genetic material (*Figure 7*).

Our findings can explain previous anecdotal reports (*Ganem et al., 2009*; *Godinho et al., 2014*; *Kuznetsova et al., 2015*; *Potapova et al., 2016*) that clones isolated after experimental inhibition of cytokinesis consisted of tetraploid cells with a 'normal' number (i.e., same number as in diploid cells) of centrosomes. Our study also shows that this pattern of centrosome number evolution after tetraploidization is common to both cancer (DLD-1) and non-cancer (RPE-1) cells. Above all, our work reveals the mechanism (*Figure 7*) by which tetraploid cells containing a normal number of centrosomes emerge. Finally, our mathematical model successfully captures tetraploid cell evolution and may be used in the future to better understand how tetraploidy contributes to tumor initiation and progression in situ.

### Tetraploidization and tumorigenesis: the case for extra chromosomes, extra centrosomes, or both as driving factors

The link between tetraploidization and tumorigenesis is supported by strong experimental evidence. Cancer genome sequencing data indicated that tetraploidization occurs at some point during the progression of a large fraction of tumors (*Zack et al., 2013*). Moreover, tetraploid

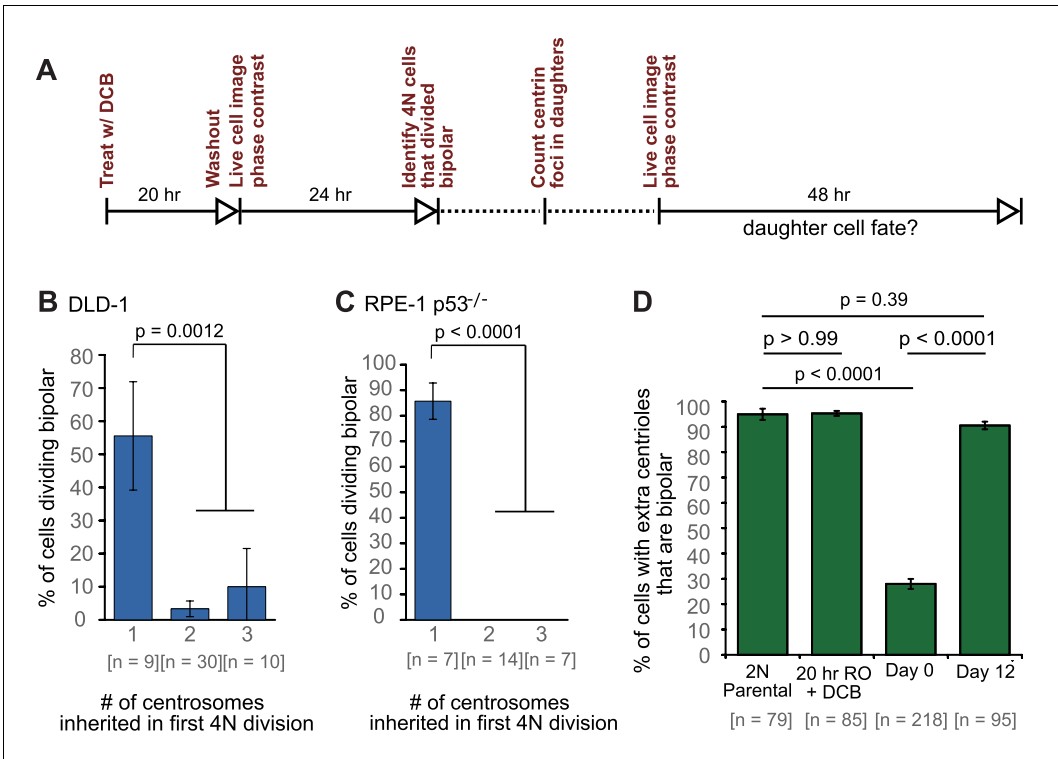

**Figure 6.** Cells that inherit a single centrosome or cells with high centrosome clustering ability are favored for continued proliferation. (**A**) Experimental design for long-term live cell imaging of cells with GFP-labeled centrin. DCB, dihydrocytochalasin B. (**B–C**) Quantification of the fraction of cells that undergo bipolar division after inheriting different numbers of centrosomes from DLD-1 (**B**) or RPE-1 p53[-/-] (**C**) tetraploid mother cells. (**D**) Fractions of bipolar ana-/telophases out of all ana-/telophases with extra centrioles in DLD-1 parental cells (2N Parental), parental cells treated with DCB while arrested in G2 (20 hr RO +DCB; to ensure that the DCB treatment did not, per se, impair centrosome clustering), tetraploid cells immediately after DCB washout (Day 0), and after twelve days of evolution (Day 12). The fraction of ana-/telophase cells with extra centrioles that display a bipolar configuration after 12 days of evolution is similar to such fraction in the parental cell line and significantly greater than such fraction in the Day 0 population. Graphs for (**B**) and (**C**) represent data collected from five and four independent experiments, respectively. Error bars represent weighted S.E.M. (weighted based on the number of cells analyzed in each experiment) and p-values were calculated by a two sided Fisher's exact test comparing the fate of cells that inherit a single centrosome to those that inherit supernumerary (2-3) centrosomes. Graph for (**D**) represents data from three independent experiments and p-values for the indicated comparisons were calculated by the two sided Fisher's exact test.

The online version of this article includes the following source data and figure supplement(s) for figure 6:

**Source data 1.** Source data for *Figure 6B–D* and *Figure 6—figure supplement 1*.

**Figure supplement 1.** Fates of daughter cells derived from bipolar mitoses with symmetric or asymmetric centrosome clustering during the first tetraploid cell division.

mouse epithelial cells were shown to be more tumorigenic than their non-tetraploid counterparts when injected in nude mice (*Fujiwara et al., 2005*; *Nguyen et al., 2009*; *Davoli and de Lange, 2012*). A popular model for how tetraploidy may promote tumorigenesis is that the extra centrosomes (which arise concomitantly with tetraploidization) contribute to cancer phenotypes (*Storchova and Pellman, 2004*). This idea is supported by the following observations: centrosome amplification is frequently observed in the pre-malignant stages of certain cancers (*Chan, 2011*; *Lopes et al., 2018*) and is observed in a large fraction of human tumors (*D'Assoro et al., 2002*; *Gustafson et al., 2000*; *Lingle et al., 1998*; *Pihan et al., 1998*; *Sato et al., 1999*), in which it correlates with poor prognosis/advanced disease stage (*Lopes et al., 2018*; *Godinho and Pellman, 2014*); extra centrosomes can promote tumorigenesis in mouse (*Levine et al., 2017*; *Serçin et al., 2016*) and enhance the invasive behavior of

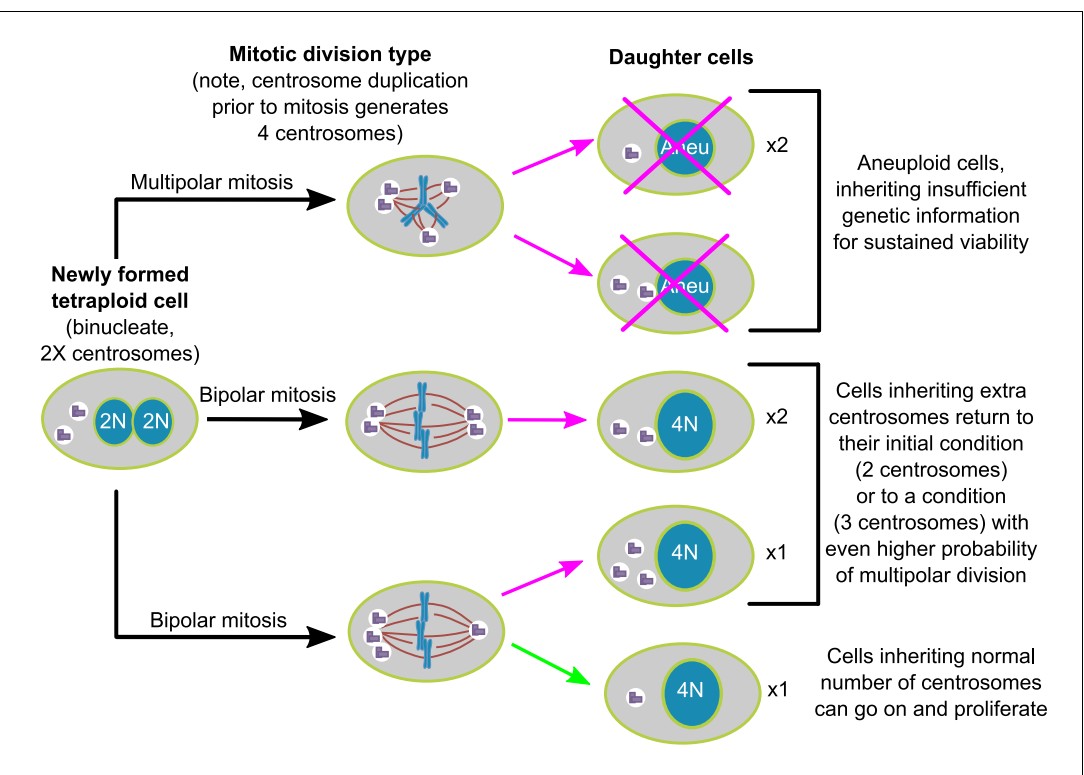

**Figure 7.** Asymmetric clustering of centrosomes defines the early evolution of tetraploid cells. The diagram illustrates the possible fates of a newly formed tetraploid cell and the mechanism by which tetraploid cells containing a normal number of centrosomes emerge.

mammary epithelial cells grown in 3D cultures (*Godinho et al., 2014*); finally, supernumerary centrosomes promote chromosome mis-attachment and mis-segregation (*Ganem et al., 2009*; *Silkworth et al., 2009*), leading to chromosomal instability, a hallmark of cancer believed to drive tumor evolution (*Targa and Rancati, 2018*). Together, these observations indicate that extra centrosomes are likely to contribute to tumor initiation and/or progression.

Our observation that extra centrosomes gained through tetraploidization are quickly lost raises the possibility that tetraploidy may drive tumorigenesis by means other than the acquisition of extra centrosomes. Indeed, a number of studies and observations suggest that tetraploidy per se may promote the emergence of cancer phenotypes. For instance, tetraploidy was shown to increase tolerance for genomic changes, leading to the rapid evolution of complex genomes (*Dewhurst et al., 2014*), as seen in cancer, and many tetraploid cells show increased chromosomal instability compared to diploid cells, even when no extra centrosomes are present (*Kuznetsova et al., 2015*). Additionally, polyploid cells were shown to be more resistant than their diploid counterparts to oxidative stress, genotoxic insult, irradiation, and certain chemotherapeutic drugs (*Kuznetsova et al., 2015*; *Ianzini et al., 2009*; *Illidge et al., 2000*). Lastly, in cancer patients, genome-doubling in early stage tumors was shown to correlate with poor relapse-free survival (*Dewhurst et al., 2014*), although centrosome number was not examined in these patients.

In light of our findings, one could imagine that in certain instances, cells experiencing a genome doubling event may initially lose their extra centrosomes and then re-acquire them at a later time, depending on additional factors. At least one example in the literature provides evidence for such a series of events. In Barrett's esophagus, a pre-malignant condition that predisposes to esophageal cancer (*Cameron et al., 1985*; *Hameeteman et al., 1989*; *Hvid-Jensen et al., 2011*), accumulation of 4N cells has been shown to occur as the tissue transitions to metaplasia (*Galipeau et al., 1996*). A study on centrosome status in Barrett's esophagus reported centrosome amplification prior to the transition to metaplasia (*Segat et al., 2010*), corresponding to the time when tetraploid cells accumulate (*Galipeau et al., 1996*), but also noted that the frequency of supernumerary centrosomes

decreased with progression to metaplasia and neoplasia (*Segat et al., 2010*). Similarly, another study found an increase in centrosome amplification followed by a decrease during the progression from Barrett's Esophagus to adenocarcinoma (*Lopes et al., 2018*). These results closely mirror the dynamics of evolution seen in our study and illustrate that extra centrosomes can be present early in tumor development (around the time when tetraploidy appears) but subsequently be lost. Therefore, while tetraploidy and supernumerary centrosomes are both individually linked with tumorigenesis, the link between tetraploidy, extra centrosomes, and disease progression may be less direct than conventionally thought.

Tetraploidization is intimately linked with the birth of extra centrosomes; however, tetraploidization may not lead to stable acquisition of supernumerary centrosomes unless (i) specific cellular/genetic changes have occurred to allow the cell to maintain its extra centrosomes and/or (ii) certain conditions in the tissue microenvironment exist that favor or necessitate the presence of extra centrosomes. Indeed, there is evidence that clustering of extra centrosomes into a bipolar configuration can be influenced by a number of cell intrinsic and extrinsic factors, and such factors may be important for determining the fraction of cells that retain extra centrosomes. Among cellular factors, the nonessential motor protein KIFC1 (also known as HSET), the epithelial cell protein E-cadherin, spindle assembly checkpoint components, the chromosome passenger complex, the NDC80 complex, and the augmin complex have been shown to affect (either positively or negatively) centrosome clustering efficiency (*Kwon et al., 2008*; *Leber et al., 2010*; *Quintyne et al., 2005*; *Rhys et al., 2018*; *Sabino et al., 2015*). Extracellular factors, such as geometric constraints imposed by the environment, have also been shown to alter centrosome clustering (*Kwon et al., 2008*). Alternatively, if tetraploidization occurs under circumstances that do not favor retention of extra centrosomes, the tetraploid cells may initially lose their extra centrosomes and then re-acquire them at a later time, as a result of genome instability, which may lead to the non-stoichiometric production of proteins involved in centrosome duplication. Thus, the evolutionary pattern that newly formed tetraploid cells will follow may vary depending on many factors, including genetic background, functional requirements in a given tissue/organ, or a variety of extracellular physical and physiological factors. All these potential factors could explain the high rates of extra centrosomes in certain tumors and animal models (*Levine et al., 2017*; *Serçin et al., 2016*).

# Materials and methods

## Key resources table

| Reagent type (species) or resource | Designation | Source or reference | Identifiers | Additional information |
|---|---|---|---|---|
| Strain, strain background (*Escherichia coli*) | NEB 5α | New England Biolabs, Inc | Cat# C2988J | Chemically competent cells |
| Strain, strain background (*Escherichia coli*) | NEB Stable | New England Biolabs, Inc | Cat# C3040I | Chemically competent cells |
| Cell line (*Homo sapiens*) | DLD-1 | ATCC | CCL-221 RRID:CVCL_0248 | |
| Cell line (*Homo sapiens*) | hTERT RPE-1 p53$^{-/-}$ | Reference 35 | | |
| Cell line (*Homo sapiens*) | DLD-1 GFP-CETN2 RFP-H2B | This study | | Cimini lab, see text for details |
| Cell line (*Homo-sapiens*) | hTERT RPE-1 p53$^{-/-}$GFP-CETN2 RFP-H2B | This study | | Cimini lab, see text for details |
| Cell line (*Homo-sapiens*) | GP-293 | Clontech | Cat # 631458 RRID:CVCL_WI48 | Viral packaging cell line |

*Continued on next page*

*Continued*

| Reagent type (species) or resource | Designation | Source or reference | Identifiers | Additional information |
|---|---|---|---|---|
| Antibody | anti-Centrin 3 clone 3E6 (Mouse monoclonal) | Abnova | Cat # H00001070-M01 RRID:AB_464016 | IF(1:100) |
| Antibody | anti-geminin EPR14637 (Rabbit monoclonal) | Abcam | Cat# ab195047 RRID:AB_2832993 | IF(1:100) |
| Antibody | anti-α-tubulin (Rabbit polyclonal) | Abcam | Cat#: ab18251 RRID:AB_2210057 | IF(1:250) |
| Other | DAPI stain | Invitrogen | D1306 | (300 nM) |
| Recombinant DNA reagent | GFP-CETN2 pLNCX2 (plasmid) | This paper | | G418 selection. Cimini lab, see text for details |
| Recombinant DNA reagent | RFP-H2B pBABE (plasmid) | Neil Ganem (Boston University) | | Puromycin selection |
| Recombinant DNA reagent | GFP-Centrin 2 pLL3.7 | Tim Stearns (Standford University) | | Origin of GFP-centrin 2 gene for retroviral vector |
| Sequence-based reagent | GfpCetn_F | This paper | PCR primers | CAATAAAGCGGC CGCATGGTGAGC AAGGGCGAG GAGCTGT |
| Sequence-based reagent | GfpCetn_R | This paper | PCR Primers | GGACTGGTGGTCT GCGTCGACTTAATA GAGGCTGGTCTT TTTCATG |
| Chemical compound, drug | Dihydrocytochalasin B | Sigma Aldrich | D1641 | (1.5 µg/ml) |
| Chemical compound, drug | Colcemid | Invitrogen | Cat # 501003406 | (50 ng/ml) |
| Chemical compound, drug | RO-3306 | Sigma Aldrich | SML0569 | (9 µM) |
| Software, algorithm | NIS elements | Nikon Instruments, Inc | RRID:SCR_014329 | AR 4.60.00 |
| Software, algorithm | FIJI | *Reference 65* | RRID:SCR_002285 | |
| Software, algorithm | MATLAB | MathWorks | RRID:SCR_001622 | R2018b |

## Experimental approaches

### Cell lines and culture conditions

DLD-1 cells (ATCC CCL-221) were purchased from the American Type Culture Collection (ATCC, Manassas, VA). The hTERT immortalized RPE-1 p53$^{-/-}$ cell line (*Izquierdo et al., 2014*) (referenced throughout the manuscript as RPE-1 p53$^{-/-}$) was a gift from Dr. Meng-Fu Bryan Tsou (Memorial Sloan Kettering Cancer Center). Both the DLD-1 and the original hTERT RPE-1 cell lines originated from ATCC. The company provides certification. Potential mycoplasma infection was monitored regularly (no less than once every three weeks) by DNA staining (DAPI) and any cell batch with suspected mycoplasma infection was discarded. DLD-1 cells were cultured according to ATCC recommendations in RPMI 1640 medium with ATCC modification (Thermo

Fisher Scientific – Gibco, CA, USA) supplemented with 10% fetal bovine serum (FBS; Thermo Fisher Scientific – Gibco, CA, USA) and 1% antibiotic-antimycotic (Thermo Fisher Scientific – Gibco, CA, USA). RPE-1 p53$^{-/-}$ cells were cultured according to the ATCC recommendations for hTERT-immortalized RPE-1 cells in 1:1 mixture of DMEM/F-12 with HEPES (Thermo Fisher Scientific – Gibco, CA, USA) also supplemented with 10% FBS and 1% antibiotic-antimycotic. All cells were grown on tissue culture polystyrene flasks (Corning, Tewksbury, MA) and were maintained in a humidified incubator at 37°C and 5% $CO_2$.

Tetraploid DLD-1 and RPE-1 p53$^{-/-}$ cells were generated by treating diploid cell cultures with 1.5 µg/mL dihydrocytochalasin B (DCB; Sigma Aldrich, Saint Louis, MO) for 20 hr. For live cell experiments, cells were washed out (4 times with 1X PBS) into imaging medium and immediately taken to the microscope for imaging following.

## Generating virally transduced cell lines

The GFP-Centrin 2 gene was PCR amplified from a modified pLL3.7 plasmid with unknown selection (gift of Tim Stearns, Stanford University), using forward and reverse primers designed to match the two termini of the fusion protein. The forward and reverse primers used (including restriction sites for NOTI and SALI and terminal non-sense nucleotides) were (with start and stop codons underlined):

(F)CAATAAAGCGGCCGCATGGTGAGCAAGGGCGAGGAGCTGT and
(R)GGACTGGTGGTCTGCGTCGACTTAATAGAGGCTGGTCTTTTTCATG.

Cleaned PCR product was ligated into the pLNXC2 retroviral expression vector by directional cloning using NOTI and SALI restriction enzymes. The presence of GFP-centrin 2 gene in plasmid DNA was confirmed by restriction digests visualized on DNA gels and via transient transfection into RPE-1 p53$^{-/-}$ cells to confirm centrosomal localization. GFP-Centrin expressing DLD-1 and RPE-1 p53$^{-/-}$ cells were generated by transduction with retroviral particles. GP-293 cells containing retroviral *gag* and *pol* genes (ClonTech Laboratories Inc, Mountain View, CA #631458) were co-transfected with the expression vector and the pVSV-G plasmid (Addgene, Cambridge, MA). 48 hr after transfection, supernatant was collected, filtered through a 0.45 µm pore (GD/X sterile 0.45 µm CA filter, GE Whatman PLC, Pittsburgh, PA), mixed with polybrene (Sigma-Aldrich, Saint Louis, MO) at a final concentration of 10 µg/ml, and added directly to the cells. After 24 hr, cell medium was replaced with fresh culture media. Starting 72 hr after viral transduction, transduced cells were selected with with G418 at a concentration of 500 µg/ml until negative control cells (untransduced cells treated with the same concentration of antibiotic) were dead, or approximately two weeks.

Cells co-expressing RFP-H2B were generated by further transducing GFP-Centrin 2 expressing cells, via the protocol described previously, using a pBABE retroviral plasmid containing RFP-H2B and a puromycin selection gene (gift from Neil Ganem, Boston University). Transduced cells were selected with puromycin at a concentration of 5 µg/ml (RPE-1 p53$^{-/-}$) or 3.8 µg/ml (DLD-1).

## Phase contrast live cell microscopy

For live-cell experiments, all cells were grown on MatTek glass bottom dishes with No. 1.5 glass (MatTek Corporation, Ashland, MA). At the time of imaging, cell medium was replaced with L-15 medium supplemented with 4.5 g/l glucose (high glucose). All live cell experiments were performed on a Nikon Eclipse Ti inverted microscope (Nikon instruments Inc, NY, USA) equipped with phase-contrast trans-illumination, transmitted light shutter, ProScan automated stage (Prior Scientific, Cambridge, UK), CoolSNAP HQ2 CCD camera (Photometrics, AZ, USA), Lumen200PRO light source (Prior Scientific, Cambridge, UK), and a temperature and humidity controlled incubator (Tokai Hit, Japan). For 24 hr and 72 hr live cell phase contrast videos, images were acquired every 6 min through a 20X/0.3 NA A Plan corrected phase contrast objective for the duration of the experiment. Time-lapse videos were analyzed using NIS Elements AR software (Nikon Instruments Inc, NY, USA) to determine the nature of division (bipolar, tripolar, tetrapolar) at anaphase and the subsequent number of daughter cells formed after cytokinesis.

## Time course experimental procedure

Time course (12 day) experiments were performed by seeding all cells needed for the first two time points (day 0 and day 2) along with a flask designated for propagating the experiment. For each

replicate for DLD-1 cells, this included T-25 flasks seeded with $1 \times 10^6$ (day 0 metaphase spreads) and $5 \times 10^5$ (day 2 metaphase spreads), a T-75 flask with $1 \times 10^6$ cells, and acid-washed coverslips inside 35 mm Petri dishes with $2.5 \times 10^5$ (day 0) and $1 \times 10^5$ (day 2) cells for combined centrin/geminin immunostaining. On day 2, the T-75 flask was used to seed cells for the next two time points as follows: two T-25 flasks (metaphase spreads), one T-75 flask (propagating), and coverslips (centrin/geminin immunostaining). This was repeated for the entire 12 day period. The experiment was designed in the same way for RPE-1 p53$^{-/-}$ cells, but cell counts were as follows: T-25 flasks seeded at $1 \times 10^6$ cells (earlier time point, e.g. day 0) and $5 \times 10^5$ (later time point, e.g. day 2); T-75 seeded at $1.5 \times 10^6$ cells; coverslips seeded at $1.25 \times 10^5$ (earlier time point) and $8.5 \times 10^4$ cells (later time point).

## Chromosome spread preparation and analysis

Cell cultures were grown in T-25 flasks as described in the previous section. For chromosome spread preparation, cells were incubated in their respective medium containing 50 ng/ml colcemid (Invitrogen – Karyomax, Waltham, MA) at 37°C for 5 hr to enrich for mitotically arrested cells. The cells were then collected by trypsinization and centrifuged at 1000 rpm for 5 min. Pre-warmed (37°C) hypotonic solution (0.075 M KCl) was added drop-wise to the cell pellet and incubated for 18 (DLD-1 cells) or 15 (RPE-1 p53$^{-/-}$ cells) minutes at 37°C. Several drops of freshly prepared fixative (3:1 methanol:glacial acetic acid) were added before centrifugation at 1000 rpm for 5 min. Supernatant was aspirated, fixative was added dropwise, and the cell suspension was again centrifuged at 1000 rpm for 5 min. The fixation step was repeated two more times and fixed cells were finally dropped on microscope slides. For RPE-1 p53$^{-/-}$ cells, a homemade humidity chamber constructed from PVC piping, plastic sheeting, and a nebulizer was used when drying slides to improve chromosome spread quality (effect of humidity on chromosome spread quality was described previously *Deng et al., 2003*). Chromosome spreads were then stained with 300 nM DAPI (Thermo Fisher Scientific – Invitrogen, Waltham, MA) for 10 min. DAPI-stained slides were mounted with an antifade solution containing 90% glycerol and 0.5% N-propyl gallate and sealed under a $22 \times 50$ mm coverslip (Corning Incorporated, Corning, NY) with nail polish. For chromosome counting, images of DAPI-stained chromosome spreads were acquired with the Nikon Eclipse Ti inverted microscope setup described in an earlier section. Images were acquired using either a 60X/1.4 NA or a 100X/1.4 NA Plan-Apochromatic phase contrast objective. After image acquisition, chromosome spreads were visualized in gray scale and chromosomes were counted using the count function in NIS elements.

## Cell death assays

To measure cell death, $5 \times 10^4$ cells were plated in each of three wells of a 6-well plate and $1 \times 10^6$ cells were plated in a T-25 flask. The following day, cells were treated with 1.5 µg/ml DCB for 20 hr. After 20 hr, day 0 cells' supernatant was collected, while the adherent cells were washed (3 times using PBS) and harvested in trypsin. The supernatant, all the washes, and the cell suspension were collected in the same tube, centrifuged, and re-suspended in 400 µl PBS for counting. Viable cells were differentiated from dead cells by trypan blue exclusion. The numbers of living and dead cells were counted and the fraction of dead cells out of the total number of cells was calculated. Cell counting was performed on days 0, 1, 2, and every 2 days for the remainder of the 12 day period (with new wells being seeded from T-25 flasks on day 2). Cell culture medium was changed 24 hr before counting each day in order to only count cells that died within a 24 hr period.

## Immunofluorescence staining, image acquisition and data analysis

For centrin and geminin immunostaining, cells were grown on sterilized acid-washed glass coverslips inside 35 mm Petri dishes. Cells were fixed in freshly prepared 4% paraformaldehyde in PHEM buffer (60 mM Pipes, 25 mM HEPES, 10 mM EGTA, 2 mM MgSO4, pH 7.0) for 20 min at room temperature and then permeabilized for 10 min at room temperature in PHEM buffer containing 0.1% Triton-X 100. Following fixation and permeabilization, cells were washed three times with PBS and then blocked with 20% boiled goat serum (BGS) for 1 hr at room temperature. Cells were then incubated at 4°C overnight with primary antibodies diluted in 10% BGS. Next, cells were washed in PBS-T (PBS with 0.05% Tween 20) three times, and incubated at room temperature for 45 min with secondary

antibodies diluted in 10% BGS. Cells were then washed four times with PBS-T, stained with DAPI (300 nM, Thermo Fisher Scientific – Invitrogen, Waltham, MA) for 5 min, and coverslips were mounted on microscope slides in an antifade solution containing 90% glycerol and 0.5% N-propyl gallate. For centrin/α-tubulin immunostaining, cells were washed in 1X PBS three times and fixed/ permeabilized in 100% methanol for 10 min. After permeabilization, fixed cells were treated as described above for centrin/geminin staining. Primary antibodies were diluted as follows: rabbit anti-geminin (Abcam, Cambridge, MA), 1:100; mouse anti-centrin (Abnova, Zhongli, Taiwan), 1:100; rabbit anti-α-tubulin (Abcam, Cambridge, MA), 1:250. Secondary antibodies were diluted as follows: Rhodamine Red-X goat anti-rabbit (Jackson ImmunoResearch Laboratories, Inc, PA, USA), 1:100; Alexa 488 goat anti-mouse (Molecular Probes, Life Technologies, CA, USA), 1:200.

Centrin-stained samples were analyzed on a Nikon Eclipse TE2000 inverted microscope equipped with a 100X/1.4 NA Plan-Apochromatic phase contrast objective lens, motorized Pro-Scan stage (Prior Scientific, Cambridge, UK), appropriate filter sets, and an XCITE 120Q light source (Excelitas Technologies, Waltham, MA, USA). Analysis was performed visually in both interphase cells and mitotic cells. The number of centrin dots was counted in cells that were determined to be in mitosis by DAPI staining. Mitotic cells with four centrin dots (i.e., two dots corresponding to each centrosome of a bipolar spindle) were categorized as normal; mitotic cells with greater than four dots were categorized as possessing supernumerary centrosomes. Interphase analysis was performed in G1/G0 cells, as determined by absence of nuclear geminin staining (*McGarry and Kirschner, 1998*). G0/G1 cells with two adjacent centrin dots (corresponding to a single centrosome) were classified as normal, whereas cells with greater than two centrin dots were classified as possessing supernumerary centrosomes. For centrosome clustering analysis (*Figure 5A–C*), bipolar metaphase, anaphase, or telophase cells were analyzed for the number of centrin dots present at respective spindle poles. For analysis of the fraction of fixed cells undergoing bipolar vs. multipolar division (*Figure 1—figure supplement 1*, *Figure 6D*), ana-/telophase cells stained with centrin and α-tubulin were analyzed for polarity (α-tubulin staining) and the presence or absence of centrioles at each spindle pole. To ensure that DCB treatment did not alter the polarity of mitotic cells, we co-treated cells with DCB and the CDK1 inhibitor RO-3306 (which causes a robust G2 arrest) for 20 hr, then washed both drugs out, waited for 1 hr for cells to proceed into mitosis, then fixed, stained and analyzed the relative proportion of bipolar and multipolar ana-/telophases in those cells naturally harboring supernumerary centrosomes. Representative z-stack image examples were acquired on the Nikon Eclipse Ti inverted microscope setup described in an earlier section. Images were acquired using either a 60X/1.4 NA or a 100X/1.4 NA Plan-Apochromatic phase contrast objective and appropriate filters.

For analysis of genome distribution in bipolar and multipolar divisions, images of ana-/telophase cells were acquired with a swept field confocal system (Prairie Technologies, WI, USA) on the same Nikon Eclipse TE2000-U inverted microscope described previously (Nikon Instruments Inc, NY, USA). The microscope was equipped with a CoolSNAP HQ2 CCD camera (Photometrics, AZ, USA), a multi-band pass filter set (illumination at 405, 488, 561, and 640 nm), and an Agilent monolithic laser combiner (MLC400) controlled by a four channel acousto-optic tunable filter. Images were obtained by acquiring Z-stacks with 0.6 µm steps (Nyquist matched) so that the entire 3-D volume of the DNA was captured. Images were shading corrected using the NIS Elements shading correction function. Z-stacks were summed using the FIJI (*Schindelin et al., 2012*) sum slices function. The freehand selection tool was used to trace the signal area corresponding to an ana-/telophase chromosome cluster and the percentage of the signal intensity relative to total DNA for an ana-/telophase cell was determined. To calculate the symmetry score, the ratio between the actual fluorescence intensity percentage and the expected signal intensity percentage for an even distribution to 2 (50%), 3 (33.3%) or 4 (25%) poles (depending on the polarity of the division) was first calculated for each chromosome cluster. Then, the standard deviation of all measurements for a cell was calculated as a 'symmetry score' (ss). If a division was perfectly symmetrical, ss = 0 and any ss >0 indicates proportional increases in the asymmetry of DNA distribution to the poles.

## Live cell imaging of fluorescently labeled cells

For live cell imaging of GFP-Centrin expressing cells, imaging was performed with a 60X/1.4 NA Plan-Apochromatic phase contrast objective lens (for RPE-1 p53[-/-] cells) or a 100X/1.4 NA Plan-

Apochromatic phase contrast objective lens (for DLD-1 cells) controlled by Nikon Perfect Focus (Nikon Instruments Inc, NY, USA). In preparation for short-term live imaging of binucleate cells expressing GFP-centrin and RFP-H2B, the cells were washed out of DCB into medium containing 9 µM of the CDK1 inhibitor RO3306 to arrest cells at the G2/M transition. After 4 hr, the cells were again washed out into high glucose L-15 medium lacking phenol red. Imaging was performed by identifying individual binucleate cells in prophase or early prometaphase using RFP-H2B signal. Cells were imaged at the home Z-position in phase contrast every 4 min and the FITC channel every 4 min with asymmetrical Z-stacks defined by the home position and a range of −2.4 µm and +5.8 µm in 0.6 µm steps (14 steps). Chromosomes were imaged by phase contrast instead of fluorescence (RFP) to keep illumination, and hence photodamage, to a minimum, given that phase contrast imaging required lower exposure times than fluorescence imaging. Cells were imaged for a total of 3–4 hr. The videos were then analyzed to determine the number of centrin dots (centrioles) in the early mitotic cells and again in the resulting daughter cells after division.

For long-term cell fate experiments (*Figure 6*), GFP-Centrin expressing cells were used. Binucleate cells were imaged at 10 min intervals for 24 hr via phase contrast microscopy under a 60X/1.4 NA or 100X/1.4 NA Plan-Apochromatic phase contrast objective lens. Following this period, a number of daughter cells were selected and the number of centrioles was quickly counted for each by eye. A phase contrast image was obtained, along with asymmetric Z-stack images in the FITC channel, defined by the home position and a range of −2.4 µm and +5.8 µm in 0.6 µm steps. These daughter cells were then tracked via phase contrast microscopy at 10 min intervals for an additional 48 hr period to determine their subsequent fate.

## Modeling approaches
### Probabilistic model for karyotypic outcomes of multipolar divisions
We built the following model to evaluate the probabilities of nullisomy and/or monosomy in a cell division with $p$ poles in a $k$-ploid mother cell, that is, a cell with $k$ sets of $M$ nonhomologous chromosomes (e.g., $k = 2$, $M = 23$ for normal, diploid human cells). For simplicity, we made the following assumptions:

1. The possibility of chromosome missegregation is ignored. Sister chromatids from each chromosome are partitioned to different spindle poles and end up in different daughter cells. The chromosome partitioning is otherwise random.
2. All chromosomes are partitioned in the same way as above and independent of one another.

Due to the second assumption, the probability of an event (e.g., nullisomy, monosomy, or nullisomy/monosomy) for at least one chromosome in a daughter cell reads as *Equation 1*. Because all chromosomes are equivalent in partitioning, the probability can be expressed in terms of the probability for Chr 1 without loss of generality.

$$\begin{aligned} P(\text{event in the cell}) \ &= 1 - \prod_{m=1}^{M} [1 - P(\text{event in Chr m in the cell})] \\ &= 1 - (1 - P(\text{event in Chr 1 in the cell}))^{M} \end{aligned} \tag{1}$$

Next, we need to determine the probability of each event of interest for Chr 1, and use *Equation 1* to deduce the corresponding probability in the cell.

### Probability of nullisomy
Because sister chromatids have to be partitioned to different poles, the total number of equal ways to partition one pair of sister chromatids to $p$ poles reads as:

$$N_{1 \times 2 \to p} = \binom{p}{2} \tag{2}$$

where the bracketed expression represents the binomial coefficient.

Because sister chromatids from each chromosome are independent of each other in the partitioning, the total number of equal ways to partition $k$ pairs of sister chromatids to $p$ poles reads as:

$$N_{k \times 2 \to p} = \binom{p}{2}^k \tag{3}$$

If any given pole receives 0 chromatids (i.e., nullisomy), then the total number of equal ways to partition $k$ pairs of sister chromatids to the remaining $p$-1 poles reads as:

$$N_{k \times 2 \to p-1} = \binom{p-1}{2}^k \tag{4}$$

Hence, the probability that any given pole and the corresponding daughter cell bears a nullisomy for Chr 1 reads as:

$$P(\text{nullisomy in Chr 1 in the cell}) = \frac{N_{k \times 2 \to p-1}}{N_{k \times 2 \to p}} = \frac{\left( \frac{(p-1)!}{(p-3)!2!} \right)^k}{\left( \frac{p!}{(p-2)!2!} \right)^k} = \left( \frac{p-2}{p} \right)^k \tag{5}$$

Note that the probability in **Equation 5** is not multiplied by another factor $p$ for the number of possible poles/daughter cells, because we are looking for the probability of nullisomy of Chr 1 in a given daughter cell rather than in a given cell division.

Plugging **Equation 5** into **Equation 1** yields the probability of nullisomy in a cell.

$$P(\text{nullisomy in the cell}) = 1 - \left( 1 - \left( \frac{p-2}{p} \right)^k \right)^M \tag{6}$$

Plugging $M$ = 23, $k$ = 4, $p$ = 3 or 4 into **Equation 6** yields the results presented in **Figure 3C** (white bars). Because all chromosomes are independent of each other, the number of nullisomies in a cell follows a binomial distribution $B(M,q)$, where $q = ((p-2)/p)^k$ according to **Equation 5**. The corresponding probability distribution for $M$ = 23, $k$ = 4, $p$ = 3 or 4 is plotted in **Figure 3B** (top).

## Probability of monosomy

If any given pole receives 1 chromatid (i.e., monosomy), then the total number of equal ways to partition the chromosomes reads as:

$$N = \underbrace{k}_{\substack{\text{choose 1 chr. pair} \\ \text{(monosomic chr.)} \\ \text{out of } k \text{ pairs}}} \times \underbrace{(p-1)}_{\substack{\text{choose 1 pole out of} \\ \text{the remaining p}-1 \text{ poles} \\ \text{for the chosen chr. pair}}} \times \underbrace{N_{(k-1) \times 2 \to p-1}}_{\substack{\text{partition the remaining} \\ \text{k}-1 \text{ chr. pairs onto the} \\ \text{remaining p}-1 \text{ poles}}} = k(p-1) \binom{p-1}{2}^{k-1} \tag{7}$$

Hence, the probability that any given pole and the corresponding daughter cell bears a monosomy for Chr 1 reads as:

$$P(\text{monosomy in Chr 1 in the cell}) = \frac{N}{N_{k \times 2 \to p}} = \frac{k(p-1) \left( \frac{(p-1)!}{(p-3)!2!} \right)^{k-1}}{\left( \frac{p!}{(p-2)!2!} \right)^k} = \frac{2k(p-2)^{k-1}}{p^k} \tag{8}$$

Plugging **Equation 8** into **Equation 1** yields the probability of monosomy in a cell.

$$P(\text{monosomy in the cell}) = 1 - \left( 1 - \frac{2k(p-2)^{k-1}}{p^k} \right)^M \tag{9}$$

Plugging $M$ = 23, $k$ = 4, $p$ = 3 or 4 into **Equation 9** yields the results presented in **Figure 3C** (grey bars). The number of monosomies in a cell follows a binomial distribution $B(M,q)$, where $q = 2k(p-2)^{k-1}/p^k$ according to **Equation 8**. The corresponding probability distribution for $M$ = 23, $k$ = 4, $p$ = 3 or 4 is plotted in **Figure 3B** (bottom).

## Probability of nullisomy or monosomy

Because nullisomy and monosomy are mutually exclusive events for a given chromosome, e.g., Chr 1, the probability that any given pole and the corresponding daughter cell bears either nullisomy or monosomy for Chr 1 reads as:

$$
\begin{aligned}
&P(\text{nullisomy or monosomy in Chr 1 in the cell}) \\
&= P(\text{nullisomy in Chr 1 in the cell}) + P(\text{monosomy in Chr 1 in the cell}) \\
&= \frac{(p-2)^k + 2k(p-2)^{k-1}}{p^k}
\end{aligned}
\tag{10}
$$

Plugging *Equation 10* into *Equation 1* yields the probability of nullisomy or monosomy in a cell.

$$
P(\text{nullisomy or monosomy in the cell}) = 1 - \left(1 - \frac{(p-2)^k + 2k(p-2)^{k-1}}{p^k}\right)^M
\tag{11}
$$

Plugging $M = 23$, $k = 4$, $p = 3$ or $4$ into *Equation 11* yields the results presented in *Figure 3C* (black bars).

## Model for centrosome number evolution in a cell population

### Model I

Model I was constructed based on the following minimal assumptions about how centrosome numbers vary during cell divisions (*Figure 5—figure supplement 1A–C*). The subscripts refer to the number of centrosomes in a cell during mitosis.

1. A cell with normal centrosome number ($C_2$) undergoes normal division with probability $q$ and cytokinesis failure ($\rightarrow C_4$) with probability $1-q$;
2. A cell with double centrosome number ($C_4$) undergoes bipolar division with probability $p$ and multipolar division with probability $1-p$;
3. A bipolar division occurs in a symmetric fashion (2 $C_4$) with probability $r$ and in an asymmetric fashion ($C_2+C_6$) with probability $1-r$.
4. A multipolar division of a $C_4$ cell goes by 2 $C_2+C_4$ with probability $s$ and 4 $C_2$ with probability $1-s$;
5. A multipolar division of a $C_4$ cell in the fashion of 4 $C_2$ is fatal;
6. A multipolar division of a $C_4$ cell in the fashion of 2 $C_2+C_4$ only has $C_4$ viable (equivalent to a normal $C_4$) with probability $f$.

In addition,

1. $C_2$ cells divide with rate $b_{C2}$, and die with rate $d_{C2}$;
2. $C_4$ cells divide with rate $b_{C4}$, and die with rate $d_{C4}$;
3. $C_6$ cells divide in multipolar fashion and die (there might be a small probability of viable division, which is neglected).

Based on the cell fate depicted in *Figure 5—figure supplement 1B–C*, the population dynamics are governed by the following ODEs:

$$
\frac{dC_2}{dt} = b_{C2}(2q-1)C_2 + b_{C4}p(1-r)C_4 - d_{C2}C_2
\tag{12}
$$

$$
\frac{dC_4}{dt} = b_{C2}(1-q)C_2 + b_{C4}(2pr + (1-p)fs - 1)C_4 - d_{C4}C_4
\tag{13}
$$

$$
\frac{dC_6}{dt} = b_{C4}p(1-r)C_4 - d_{C6}C_6
\tag{14}
$$

with initial condition $C_2(0) = \alpha N$, $C_4(0) = (1-\alpha)N$, $C_6(0) = 0$. The initial condition reflects the experimental observation that the newly induced 4N cell populations usually contain a certain fraction ($\alpha$) of $C_2$ (2N) cells.

Parameter sensitivity analysis (*Figure 5—figure supplement 1D*) indicated that the final fraction of cells with extra centrosomes strongly depends on $q$, the probability of cytokinesis failure in cells

with normal centrosome number. In fact, the range of possible values for this final fraction is strongly constrained by the value of $q$, even if choice of all parameters could span a wide range (*Figure 5—figure supplement 3A*). This is because cytokinesis failure causes formation of new cells with extra centrosomes, and hence a large probability of cytokinesis failure is needed to maintain a higher fraction of these cells in the population.

## Model II (with SC cells)

In the updated model (*Figure 5—figure supplement 2A-C*), we added SC cells, which are $C_4$ cells that can effectively cluster extra centrosomes, and regularly undergo bipolar division. For this new cell type, we made the following assumptions.

1. Cytokinesis failure in cells with normal centrosome number results in SC cells with probability, $v$.
2. SC cells divide symmetrically (SC+SC) with a probability, $r_S$. Otherwise, they divide asymmetrically ($C_2$+$C_6$).
3. SC cells have the same division and death rates as cells with normal centrosome number, because they are supposedly stable.

Based on the cell fate depicted in *Figure 5—figure supplementv 2B-C*, the population dynamics are governed by the following ODEs:

$$\frac{dC_2}{dt} = b_{C2}(2q-1)C_2 + b_{C4}p(1-r)C_4 + b_{C2}(1-r_S)SC - d_{C2}C_2 \tag{15}$$

$$\frac{dC_4}{dt} = b_{C2}(1-q)(1-v)C_2 + b_{C4}(2pr+(1-p)fs-1)C_4 - d_{C4}C_4 \tag{16}$$

$$\frac{dSC}{dt} = b_{C2}(1-q)vC_2 + b_{C2}(2r_S-1)SC - d_{C2}SC \tag{17}$$

$$\frac{dC_6}{dt} = b_{C4}p(1-r)C_4 + b_{C2}(1-r_S)SC - d_{C6}C_6 \tag{18}$$

Parameter sensitivity analysis (*Figure 5—figure supplement 2D*) indicated that, based on Model II, the final fraction of cells with extra centrosomes is most sensitive to $r_S$, the probability of symmetric division in SC cells, followed by $q$, the probability of cytokinesis failure in $C_2$ cells, and $v$, the probability of getting SC cells upon cytokinesis failure. While Model I showed a strong constraint on $q$ (*Figure 5—figure supplement 3A*), the strength of this constraint is relaxed in Model II (*Figure 5—figure supplement 3B*). In Model II, the major constraint is shifted to $r_S$ (*Figure 5—figure supplement 3C*), because asymmetric division (with probability 1- $r_S$) controls the conversion of SC cells back to $C_2$ cells. Nevertheless, ~90% probability of symmetric division is sufficient to maintain 20% cells with extra centrosomes in the steady state population.

## Steady state of cell fractions

When the cell division rate is sufficiently large compared to cell death rate in the models, the number of cells in each type will increase infinitely (*Figure 5—figure supplement 4*, left column). This case does reflect the experiments, in which the cell cultures were regularly sampled and re-populated on fresh medium, effectively creating a finite sample of the unlimited population growth. Although the total population grows infinitely, the fractions of each cell type approach fixed steady state values (*Figure 5—figure supplement 4*, right column). In fact, the steady state fraction of each cell type can be analytically solved as shown below.

Systems of homogenous linear ODE equations like *Equations 12-14* and *Equations 15-18* can be written in a vector form as

$$\frac{d\mathbf{X}}{dt} = \mathbf{P} \cdot \mathbf{X} \tag{19}$$

where $\mathbf{X} = (X_1, X_2, \ldots, X_N)$ is the list of variables.

The coefficient matrix, **P**, has the rate constants as entries. For Model I governed by *Equations 12-14*,

$$\mathbf{P} = \begin{bmatrix} b_{C2}(2q-1) - d_{C2} & b_{C4}p(1-r) & 0 \\ b_{C2}(1-q) & b_{C4}(2pr + (1-p)fs - 1) - d_{C4} & 0 \\ 0 & b_{C4}p(1-r) & -d_{C6} \end{bmatrix} \tag{20}$$

Likewise, for Model II governed by *Equations 15-18*,

$$\mathbf{P} = \begin{bmatrix} b_{C2}(2q-1) - d_{C2} & b_{C4}p(1-r) & b_{C2}(1-r_S) & 0 \\ b_{C2}(1-q)(1-v) & b_{C4}(2pr + (1-p)fs - 1) - d_{C4} & 0 & 0 \\ b_{C2}(1-q)v & 0 & b_{C2}(2r_S - 1) - d_{C2} & 0 \\ 0 & b_{C4}p(1-r) & b_{C2}(1-r_S) & -d_{C6} \end{bmatrix} \tag{21}$$

If $\det(\mathbf{P}) \neq 0$, then *Equation 19* only has the trivial steady state where all variables equal zero. This trivial steady state is unstable if the overall proliferation rate is larger than the overall death rate. In other words, the whole cell population is expected to increase infinitely. Although the total population grows infinitely, the fraction of each cell type in the population could reach a steady state. To address this question via modeling, one can rewrite *Equation 19* in terms of the fraction of each cell, that is,

$$f_i := \frac{X_i}{\sum_j X_j} \tag{22}$$

Hence,

$$\begin{aligned} \frac{df_i}{dt} &= \frac{X_i'}{\sum_j X_j} - \frac{X_i \sum_j X_j'}{\left(\sum_j X_j\right)^2} \\ &= \frac{\sum_j P_{ij} X_j}{\sum_j X_j} - \frac{X_i}{\sum_j X_j} \frac{\sum_j \left(\sum_k P_{kj}\right) X_j}{\sum_j X_j} \\ &= \sum_j P_{ij} f_j - f_i \sum_j \left(\sum_k P_{kj}\right) f_j \end{aligned} \tag{23}$$

*Equation 23* can be rewritten in vector format as

$$\frac{d\mathbf{f}}{dt} = \mathbf{P} \cdot \mathbf{f} - C\mathbf{f} \tag{24}$$

where $\mathbf{f} = (f_1, f_2, \ldots, f_N)$ and $C(t) = \sum_j \left(\sum_k P_{kj}\right) f_j(t)$.

Because $C(t)$ is a scalar function of time, at the steady state of *Equation 24*, $C(t)$ approaches a constant, that is, $C(t) \xrightarrow{t \to \infty} C^\infty$. In other words, the steady state of *Equation 24* is found when

$$\mathbf{P} \cdot \mathbf{f} = C^\infty \mathbf{f} \tag{25}$$

Hence, the steady state solution of *Equation 24* is an eigenvector of the coefficient matrix, **P**, normalized by the constraint, $\sum_i f_i = 1$. $C^\infty$ equals the corresponding eigenvalue of **P**. We show in the following that $C^\infty$ is in fact the largest eigenvalue of **P**.

**Theorem 1:** The steady state solution of *Equation 24* is given by the normalized eigenvector associated with the largest eigenvalue of the coefficient matrix, **P**, with the normalization condition, $\sum_i f_i = 1$.

Heuristic proof:

At $t \to \infty$, the solution to *Equation 24* approaches the solution to *Equation 26*.

$$\frac{d\mathbf{g}}{dt} = \mathbf{P} \cdot \mathbf{g} - C^\infty \mathbf{g} \tag{26}$$

The solution of *Equation 26* reads

$$g_i(t) = \sum_k q_{ik} e^{\lambda_k t} \tag{27}$$

where $\lambda_k$'s are eigenvalues of the matrix $\mathbf{Q} = \mathbf{P} - C^\infty \mathbf{I}$, and $\mathbf{I}$ is the identity matrix.

At $t \to \infty$, *Equation 27* is dominated by the term with the largest eigenvalue, that is,

$$g_i(t) \overset{t\to\infty}{\to} q_{i0} e^{\lambda_{\max} t} \tag{28}$$

A nonzero steady state solution requires $\lambda_{\max} = 0$. Note that the eigenvalues of $\mathbf{P}$ have one-to-one correspondence with the eigenvalues of $\mathbf{Q}$. For each eigenvalue of $\mathbf{Q}$, $\lambda_k$, $\lambda_k + C^\infty$ is an eigenvalue of $\mathbf{P}$. Because the largest eigenvalue of $\mathbf{Q}$ is 0, the largest eigenvalue of $\mathbf{P}$ is $C^\infty$. The normalization constraint follows from the definition of fractions in *Equation 22*.

Based on Theorem 1, the steady state fractions of each cell type in the model can be obtained by computing the normalized eigenvector associated with the largest eigenvalue of the coefficient matrix, $\mathbf{P}$, which can be easily done using a computation software, for example, MATLAB.

## Acknowledgements

We would like to thank the labs of Neil Ganem and Tim Stearns for reagents and Meng-Fu Bryan Tsou for the RPE-1 p53$^{-/-}$ cell line. We further acknowledge members of the Cimini, Chen, and Hauf labs for helpful discussion. Finally, we would like to thank Dr. Silke Hauf for critical reading of the manuscript.

## Additional information

### Funding

| Funder | Grant reference number | Author |
| --- | --- | --- |
| Virginia Tech | College of Science Dean's Discovery Fund | Daniela Cimini |
| Virginia Tech | Fralin Life Sciences Institute Discretionary funds | Daniela Cimini |
| Virginia Tech | ICTAS Center for Engineered Health Seed funding | Daniela Cimini |
| National Science Foundation | MCB-1517506 | Daniela Cimini |
| Virginia Tech | BIOTRANS IGEP | Nicolaas C Baudoin |

The funders had no role in study design, data collection and interpretation, or the decision to submit the work for publication.

### Author contributions

Nicolaas C Baudoin, Conceptualization, Data curation, Formal analysis, Investigation, Methodology, Writing - original draft, Writing - review and editing, Performed most of the experimental work and contributed to the chromosome segregation modeling work, study design, data analysis, figure preparation, data interpretation and writing of the manuscript; Joshua M Nicholson, Conceptualization, Formal analysis, Investigation, Writing - review and editing, Contributed to study design, data collection, data analysis, and approved final version of the manuscript; Kimberly Soto, Olga Martin, Formal analysis, Writing - review and editing, Performed analysis of experimental data and approved final version of the manuscript; Jing Chen, Conceptualization, Resources, Data curation, Formal analysis, Methodology, Writing - original draft, Writing - review and editing, Contributed to the chromosome segregation modeling work, generated the model for centrosome number evolution and

associated modeling results, contributed to study design, figure preparation, data interpretation, writing of the manuscript, and provided computational resources; Daniela Cimini, Conceptualization, Resources, Supervision, Funding acquisition, Investigation, Writing - original draft, Project administration, Writing - review and editing, Contributed to study design, figure preparation, data interpretation, and writing of the manuscript and provided reagents, equipment, and other resources

## Author ORCIDs
Nicolaas C Baudoin (ID) https://orcid.org/0000-0003-3316-5293
Jing Chen (ID) https://orcid.org/0000-0001-6321-0505
Daniela Cimini (ID) https://orcid.org/0000-0002-4082-4894

## Decision letter and Author response
Decision letter https://doi.org/10.7554/eLife.54565.sa1
Author response https://doi.org/10.7554/eLife.54565.sa2

# Additional files

## Supplementary files
• Transparent reporting form

## Data availability
All data generated during the study are provided in clearly labeled source data files in excel format.

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
