## [Decision Letter]

**Acceptance summary:**

Your observation that asymmetric centrosome segregation occurs in cells with multiple centrosomes to derive some cells with normal centrosome numbers, and that this provides a fitness advantage that will allow these cells to dominate the population over time, provides a mechanism to explain how tetraploid cells with multiple centrosomes derived from cytokinesis failure can revert to normal centrosome numbers. Your proposal that some cells in an initial population have an enhanced ability to cluster centrosomes could be important to understand how tumours evolve. Thus your study is an important contribution to the literature.

**Decision letter after peer review:**

Thank you for submitting your article "Asymmetric clustering of centrosomes defines the early evolution of tetraploid cells" for consideration by *eLife*. Your article has been reviewed by three peer reviewers, and the evaluation has been overseen by Jon Pines as Reviewing Editor and Anna Akhmanova as the Senior Editor. The reviewers have opted to remain anonymous.

The reviewers have discussed the reviews with one another and the Reviewing Editor has drafted this decision to help you prepare a revised submission.

Summary:

In this manuscript, Baudoin et al. propose that the reversion to normal centrosome number following cytokinesis failure results from the difference in fitness between daughter cells after mitosis with asymmetric centrosome clustering. The authors show that some daughter cells stochastically receive a single centrosome following mitosis, and these cells are more likely to continue proliferating. Finally, through computational analysis, the authors propose the existence of a population of “super-clustering” cells that can stably maintain additional centrosomes through a heightened ability to cluster centrosomes in mitosis.

This study nicely confirms the logical conclusion that supernumerary centrosomes cause a fitness disadvantage and that imperfect clustering of centrosomes in mitosis stochastically generates cells with a normal centrosome content. Due to the rapid proliferation of cells in culture, the growth advantages of cells with normal centrosome numbers quickly leads to their outgrowth within the population. The observations made by the authors in this manuscript provide a nice confirmation of what could be assumed to be the most likely mechanism by which supernumerary centrosomes are resolved in rapidly proliferating cultured cells. Overall, the study is nicely constructed, well-executed, and the data support their claims.

Essential revisions:

The main experimental support for the existence of SC cells is in Figure 7D. The data show that immediately after cytokinesis failure by cytochalasin treatment and washout, most cells fail to efficiently cluster extra centrosomes whereas 12 days later, clustering efficiency appears higher. My concern is whether this effect can be explained by an artifact from drug treatment rather than reflecting true alterations in the fraction of genetically or epigenetically distinct "SC" cells. In other words, has the transient drug treatment artificially suppressed a baseline fairly efficient process of centrosome clustering. In principle, this might occur because of small amounts of drug retained in cells, a stress response to drug treatment, and/or changes to the actin cytoskeleton that are not immediately resolved after drug treatment.

To address this concern, I suggest two experiments. First, I suggest arresting cells in G1 with serum starvation, performing the DCB treatment and washout followed by release from the arrest, and determine if DCB treatment in that cell cycle but not during cytokinesis, independent of effects on cytokinesis, affects centrosome clustering efficiency. This would either need to be done by generating cells with extra centrosomes (e.g. transient Plk4) or by restricting the analysis to the cells in the starting population that spontaneously have extra centrosomes. Second, if the evolved population truly has an enhanced capacity to cluster extra centrosomes, then it is predicted that there will be a high efficiency of centrosome clustering in the evolved cells when they are re-challenged with the DCB treatment/washout procedure.

• Figure 1, 4: The findings were based on phase-contrast live-cell imaging of mitotic tetraploid cells where the authors explained that the reasoning for high rates of multipolar mitoses during the first 24h post DCB were due to acquiring extra centrosomes upon tetraploidization followed by the subsequent formation of multipolar spindle (subsection “Newly formed tetraploid cells undergo diverse fates in their first mitotic division”). These assumptions could be more robustly shown if the authors performed live-cell fluorescent imaging on cells expressing fluorescent centrosomes and nuclei/microtubules. This way the centrosomes could be clearly tracked to their respective poles for accurate quantification as it is well documented that spindle formation can sometimes occur without the presence of centrosomes (e.g. in the case of PCM fragmentation). Furthermore, this would have consolidated the need to perform the fixed analysis of centrosomes in Figure 4. Figure 1, S1: A representative picture of what was quantified would be help interpret the data. Figure 1, S1A, S1B could be combined into a single graph, averaging the cell data in each respective case to allow for easy comparison.

• Figure 5: The basis of the model for centrosome evolution in Figure 5 was based on the parameters from preceding experiments. The modeling results display high similarity with the “experimental” data. This experimental data was however used in forming the mathematical model. A new set of triplicate experiments should be performed to ensure that the modeling was accurate in simulating centrosome evolution.

---

## [Author Response]

Essential revisions:The main experimental support for the existence of SC cells is in Figure 7D. The data show that immediately after cytokinesis failure by cytochalasin treatment and washout, most cells fail to efficiently cluster extra centrosomes whereas 12 days later, clustering efficiency appears higher. My concern is whether this effect can be explained by an artifact from drug treatment rather than reflecting true alterations in the fraction of genetically or epigenetically distinct "SC" cells. In other words, has the transient drug treatment artificially suppressed a baseline fairly efficient process of centrosome clustering. In principle, this might occur because of small amounts of drug retained in cells, a stress response to drug treatment, and/or changes to the actin cytoskeleton that are not immediately resolved after drug treatment.To address this concern, I suggest two experiments. First, I suggest arresting cells in G1 with serum starvation, performing the DCB treatment and washout followed by release from the arrest, and determine if DCB treatment in that cell cycle but not during cytokinesis - independent of effects on cytokinesis - affects centrosome clustering efficiency. This would either need to be done by generating cells with extra centrosomes (e.g. transient Plk4) or by restricting the analysis to the cells in the starting population that spontaneously have extra centrosomes. Second, if the evolved population truly has an enhanced capacity to cluster extra centrosomes, then it is predicted that there will be a high efficiency of centrosome clustering in the evolved cells when they are re-challenged with the DCB treatment/washout procedure.

We found that serum starvation did not effectively induce a cell cycle arrest in DLD-1 cells. Therefore, we performed a slight variation of the experiment suggested here. We used RO3306 to induce a G2 arrest. The cells were co-treated with RO3306 and DCB for 20 hrs, the drugs were then washed out, and the cells were fixed and stained after one hour. The analysis was restricted to cells with extra centrosomes, which represent cells from the parental population that already had extra centrosomes. The results from these experiments are now included in Figure 6D (formerly 7D) and show that these cells can cluster their centrosomes as efficiently as the cells with extra centrosomes in the untreated parental population. This indicates that the DCB treatment *per se* does not affect centrosome clustering efficiency.

We do not think that the second experiment suggested here would be informative and the results would likely be very difficult to interpret for the following reasons: in the evolved population, the remaining cells with extra centrosomes that display high efficiency of centrosome clustering are mainly tetraploid cells. If these cells were challenged again with DCB, they would produce octoploid cells with eight centrosomes. We do not expect cells to cluster eight centrosomes as efficiently as they would cluster four centrosomes. Moreover, the presence of eight sets of chromosomes may further hinder centrosome clustering. Therefore, the results of such an experiment would be extremely difficult to interpret and would likely be uninformative.

• Figure 1, 4: The findings were based on phase-contrast live-cell imaging of mitotic tetraploid cells where the authors explained that the reasoning for high rates of multipolar mitoses during the first 24h post DCB were due to acquiring extra centrosomes upon tetraploidization followed by the subsequent formation of multipolar spindle (subsection “Newly formed tetraploid cells undergo diverse fates in their first mitotic division”). These assumptions could be more robustly shown if the authors performed live-cell fluorescent imaging on cells expressing fluorescent centrosomes and nuclei/microtubules. This way the centrosomes could be clearly tracked to their respective poles for accurate quantification as it is well documented that spindle formation can sometimes occur without the presence of centrosomes (e.g. in the case of PCM fragmentation). Furthermore, this would have consolidated the need to perform the fixed analysis of centrosomes in Figure 4. Figure 1, S1: A representative picture of what was quantified would be help interpret the data. Figure 1, S1A, S1B could be combined into a single graph, averaging the cell data in each respective case to allow for easy comparison.

To address this point, we have now added fixed-cell data to show that multipolar division was not due to the formation of acentrosomal spindle poles or centrosome fragmentation, as a pair of centrin dots were always found at each spindle pole. Some of our live-cell imaging data from the experiments presented in what was previously Figure 6 could also be used to address this point. Therefore, we have moved the data from Figure 6 to an earlier section of the manuscript. These data and the fixed-cell data are now presented in Figure 1—figure supplement 1. And the data are discussed in subsection “Newly formed tetraploid cells undergo diverse fates in their first mitotic division”.

We have also revised Figure 1, S1 based on the suggestion. However, we have moved the discussion of these data to a later section of the Results (subsection “Highly aneuploid cells form early in the evolution of tetraploid cells, but quickly disappear from the population”) and the figure is now Figure 2—figure supplement 1.

• Figure 5: The basis of the model for centrosome evolution in Figure 5 was based on the parameters from preceding experiments. The modeling results display high similarity with the “experimental” data. This experimental data was however used in forming the mathematical model. A new set of triplicate experiments should be performed to ensure that the modeling was accurate in simulating centrosome evolution.

Performing additional time-course experiments would simply show reproducibility of our experimental results, not the validity of the model. Instead, validity of the model can be assessed by experimentally testing model assumptions and/or predictions. In our study, an important assumption of the model was that 4N cells inheriting a single centrosome would be the most likely to continue dividing in a bipolar fashion, thus maintaining a ~4N chromosome number in cells with a single centrosome. This assumption was tested in live-cell experiments, results of which are presented in Figure 6A-C (formerly Figure 7) and Figure 6—figure supplement 1. The main prediction of the model was that of the remaining cells with extra centrosomes in the evolved population, a vast majority would be represented by SC cells, which can cluster the extra centrosomes very efficiently. This prediction was tested experimentally and the results are presented in Figure 6D.